# On Proper Learnability between Average- and Worst-case Robustness

**Vinod Raman**[*]
Department of Statistics
University of Michigan
Ann Arbor, MI 48104
vkraman@umich.edu

**Unique Subedi**[*]
Department of Statistics
University of Michigan
Ann Arbor, MI 48104
subedi@umich.edu

**Ambuj Tewari**
Department of Statistics
University of Michigan
Ann Arbor, MI 48104
tewaria@umich.edu

## Abstract

Recently, Montasser et al. [2019] showed that finite VC dimension is not sufficient for *proper* adversarially robust PAC learning. In light of this hardness, there is a growing effort to study what type of relaxations to the adversarially robust PAC learning setup can enable proper learnability. In this work, we initiate the study of proper learning under relaxations of the adversarially robust loss. We give a family of robust loss relaxations under which VC classes are properly PAC learnable with sample complexity close to what one would require in the standard PAC learning setup. On the other hand, we show that for an existing and natural relaxation of the adversarially robust loss, finite VC dimension is not sufficient for proper learning. Lastly, we give new generalization guarantees for the adversarially robust empirical risk minimizer.

## 1 Introduction

As deep neural networks become increasingly ubiquitous, their susceptibility to test-time adversarial attacks has become more and more apparent. Designing learning algorithms that are robust to these test-time adversarial perturbations has garnered increasing attention by machine learning researchers and practitioners alike. Prior work on adversarially robust learning has mainly focused on learnability under the worst-case *adversarially* robust risk [Montasser et al., 2019, Attias et al., 2021, Cullina et al., 2018],

$$R_{\mathcal{U}}(h; \mathcal{D}) := \mathop{\mathbb{E}}_{(x,y)\sim\mathcal{D}} \left[ \sup_{z\in\mathcal{U}(x)} \mathbb{1}\{h(z) \neq y\} \right],$$

where $\mathcal{U}(x) \subset \mathcal{X}$ is an arbitrary but fixed perturbation set (for example $\ell_p$ balls). In practice, worst-case robustness is commonly achieved via Empirical Risk Minimization (ERM) of the adversarially robust loss or some convex surrogate [Madry et al., 2017, Wong and Kolter, 2018, Raghunathan et al., 2018, Bao et al., 2020]. However, a seminal result by Montasser et al. [2019] shows that any proper learning rule, including ERM, even when trained on an arbitrarily large number of samples, may not return a classifier with small adversarially robust risk. These high generalization gaps for the adversarially robust loss have also been observed in practice [Schmidt et al., 2018]. Even worse, empirical studies have shown that classifiers trained to achieve worst-case adversarial robustness exhibit degraded *nominal* performance [Dobriban et al., 2020, Raghunathan et al., 2019, Su et al., 2018, Tsipras et al., 2018, Yang et al., 2020, Zhang et al., 2019, Robey et al., 2022].

In light of these difficulties, there has been a recent push to study when proper learning, and more specifically, when learning via ERM is possible for achieving adversarial robustness. The ability to achieve test-time robustness via proper learning rules is important from a practical standpoint. It

---

[*]Equal contributions

37th Conference on Neural Information Processing Systems (NeurIPS 2023).

aligns better with the current approaches used in practice (e.g. (S)GD-trained deep nets), and proper learning algorithms are often simpler to implement than improper ones. In this vain, Ashtiani et al. [2022] and Bhattacharjee et al. [2022] consider adversarial robust learning in the *tolerant* setting, where the error of the learner is compared with the best achievable error with respect to a slightly *larger* perturbation set. They show that the sample complexity of tolerant robust learning can be significantly lower than the current known sample complexity for adversarially robust learning and that proper learning via ERM can be possible under certain assumptions. Additionally, Ashtiani et al. [2020] provide some sufficient conditions under which proper robust learnability of VC classes becomes possible under a PAC-type framework of semi-supervised learning. More recently, Attias et al. [2022] show that finite VC dimension is sufficient for proper semi-supervised adversarially robust PAC learning if the support of the marginal distribution on instances is known. On the other hand, Attias and Hanneke [2023] study the adversarially robust PAC learnability of real-valued functions and show that every convex function class is properly learnable.

In a different direction, several works have studied the consistency of various *surrogates* of the adversarially robust loss $\ell_{\mathcal{U}}(h, (x, y)) = \sup_{z \in \mathcal{U}(x)} \mathbb{1}\{h(z) \neq y\}$ [Awasthi et al., 2022a,b, 2023, Mao et al., 2023], while others have considered relaxing its worst-case nature [Robey et al., 2022, Li et al., 2020, 2021, Laidlaw and Feizi, 2019, Rice et al., 2021]. However, the PAC learnability of these relaxed notions of the adversarially robust loss has not been well-studied.

In this paper, we study relaxations of the worst-case adversarially robust learning setup from a learning-theoretic standpoint. We classify existing relaxations of worst-case adversarially robust learning into two approaches: one based on relaxing the loss function and the other based on relaxing the benchmark competitor. Much of the existing learning theory work studying relaxations of adversarial robustness focus on the latter approach. These works answer the question of whether proper PAC learning is feasible if the learner is evaluated against a stronger notion of robustness. In contrast, we focus on the *former* relaxation and pose the question: *can proper PAC learning be feasible if we relax the adversarially robust loss function itself?* In answering this question, we make the following main contributions:

- We show that the finiteness of the VC dimension is *not sufficient* for properly learning a natural relaxation of the adversarially robust loss proposed by Robey et al. [2022]. Our proof techniques involve constructing a VC class that is not properly learnable.
- We give a family of adversarially robust loss relaxations that interpolate between average- and worst-case robustness. For these losses, we use Rademacher complexity arguments relying on the Ledoux-Talagrand contraction to show that all VC classes are learnable via ERM.
- We extend a property implicitly appearing in margin theory (e.g., see Mohri et al. [2018, Section 5.4]), which we term "Sandwich Uniform Convergence" (SUC), to show new generalization guarantees for the adversarially robust empirical risk minimizer.

## 2 Preliminaries and Notation

Throughout this paper we let $[k]$ denote the set of integers $\{1, ..., k\}$, $\mathcal{X}$ denote an instance space, $\mathcal{Y} = \{-1, 1\}$ denote our label space, and $\mathcal{D}$ be any distribution over $\mathcal{X} \times \mathcal{Y}$. Let $\mathcal{H} \subseteq \mathcal{Y}^{\mathcal{X}}$ denote a hypothesis class mapping examples in $\mathcal{X}$ to labels in $\mathcal{Y}$.

### 2.1 Problem Setting

In the standard adversarially robust learning setting, the learner, during training time, picks a set $\mathcal{G} \subseteq \mathcal{X}^{\mathcal{X}}$ [2] of perturbation functions $g : \mathcal{X} \rightarrow \mathcal{X}$ against which they wish to be robust. At test time, an adversary intercepts the labeled example $(x, y)$, exhaustively searches over the perturbation set to find *the worst* $g \in \mathcal{G}$, and then passes the perturbed instance $g(x)$ to the learner. The learner makes a prediction $\hat{y}$ and then suffers the loss $\mathbb{1}\{\hat{y} \neq y\}$. The goal of the learner is to find a hypothesis $h \in \mathcal{Y}^{\mathcal{X}}$ that minimizes the adversarially robust risk $R_{\mathcal{G}}(h; \mathcal{D}) = \mathbb{E}_{(x,y) \sim \mathcal{D}} [\ell_{\mathcal{G}}(h, (x, y))]$ where $\ell_{\mathcal{G}}(h, (x, y)) := \sup_{g \in \mathcal{G}} \mathbb{1}\{h(g(x)) \neq y\}$ is the adversarially robust loss.

---

[2]We highlight that our use of perturbation functions $\mathcal{G}$ instead of perturbation sets $\mathcal{U}$ is without loss of generality (see Appendix A for an equivalence).

In practice, however, such a worst-case adversary may be too strong and unnatural, especially in high-dimension. Accordingly, we relax this model by considering a *lazy*, computationally-bounded adversary that is unable to exhaustively search over $\mathcal{G}$, but can randomly sample a perturbation function $g \sim \mu$ given access to a measure $\mu$ over $\mathcal{G}$. In this setup, the learner picks both a perturbation set $\mathcal{G}$ and a measure $\mu$ over $\mathcal{G}$. At test-time, the lazy adversary intercepts the labeled example $(x, y)$, *randomly samples* a perturbation function $g \sim \mu$, and then passes the perturbed instance $g(x)$ to the learner. From this perspective, the goal of the learner is to output a hypothesis such that the *probability* that the lazy adversary succeeds in sampling a bad perturbation function, for any labeled example in the support of $\mathcal{D}$, is small. In other words, the learner strives to be robust to most, but not all, perturbation functions in $\mathcal{G}$.

We highlight that the set $\mathcal{G}$ and the measure $\mu$ are fixed and chosen by the learner at training time. As a result, the measure $\mu$ does not depend on the unperturbed point $x$. Allowing the measure to depend on the unperturbed point $x$ is an interesting future direction that lies between our model and adversarial robustness. Nevertheless, fixing the measure $\mu$ is still relevant from a practical standpoint and non-trivial from a theoretical standpoint. Indeed, a popular choice for $\mathcal{G}$ and $\mu$ are $\ell_p$ balls and the uniform measure respectively. These choices have been shown experimentally to strike a better balance between robustness and nominal performance [Robey et al., 2022]. Moreover, in Section 3, we show that even when the measure $\mu$ is fixed apriori, there are learning problems for which ERM does not always work. This parallels the hardness result from Montasser et al. [2019], who prove an analogous result for adversarially robust PAC learning.

## 2.2 Probabilistically Robust Losses and Proper Learnability

Given a perturbation set $\mathcal{G}$ and measure $\mu$, we quantify the probabilistic robustness of a hypothesis $h$ on a labeled example $(x, y)$ by considering losses that are a function of $\mathbb{P}_{g \sim \mu}[h(g(x)) \neq y]$. For a labeled example $(x, y) \in \mathcal{X} \times \mathcal{Y}$, $\mathbb{P}_{g \sim \mu}[h(g(x)) \neq y]$ measures the fraction of perturbations in $\mathcal{G}$ for which the classifier $h$ is non-robust. Observe that $\mathbb{P}_{g \sim \mu}[h(g(x)) \neq y] = \frac{1 - y\mathbb{E}_{g \sim \mu}[h(g(x))]}{2}$ is an affine transformation of quantity $y\mathbb{E}_{g \sim \mu}[h(g(x))]$, the probabilistically robust *margin* of $h$ on $(x, y)$ with respect to $(\mathcal{G}, \mu)$. Thus, we focus on loss functions that operate over the margin $y\mathbb{E}_{g \sim \mu}[h(g(x))]$. One important loss function is the $\rho$-probabilistically robust loss,

$$\ell^\rho_{\mathcal{G}, \mu}(h, (x, y)) := \mathbb{1}\{\mathbb{P}_{g \sim \mu}(h(g(x)) \neq y) > \rho\},$$

where $\rho \in [0, 1)$ is selected apriori. The $\rho$-probabilistically robust loss was first introduced by Robey et al. [2022] for the case when $\mathcal{X} = \mathbb{R}^d$, $g_c(x) = x + c$, and the set of perturbations $\mathcal{G} = \{g_c : c \in \Delta\}$ for some $\Delta \subset \mathbb{R}^d$. In this paper, we generalize this loss to an arbitrary instance space $\mathcal{X}$ and perturbation set $\mathcal{G}$. As highlighted by Robey et al. [2022], this notion of robustness is desirable as it nicely interpolates between worst- and average-case robustness via an interpretable parameter $\rho$, while being more computationally tractable compared to existing relaxations.

In this work, we are primarily interested in understanding whether probabilistic relaxations of the adversarially robust loss enable *proper learning*. That is, given a hypothesis class $\mathcal{H}$, perturbation set and measure $(\mathcal{G}, \mu)$, loss function $\ell_{\mathcal{G}, \mu}(h, (x, y)) = \ell(y\mathbb{E}_{g \sim \mu}[h(g(x))])$, and labeled samples from an unknown distribution $\mathcal{D}$, our goal is to design a learning algorithm $\mathcal{A} : (\mathcal{X} \times \mathcal{Y})^* \to \mathcal{H}$ such that for any distribution $\mathcal{D}$ over $\mathcal{X} \times \mathcal{Y}$, the algorithm $\mathcal{A}$, given a sample of labeled examples from $\mathcal{D}$, finds a hypothesis $h \in \mathcal{H}$ with low risk with regards to $\ell_{\mathcal{G}, \mu}(h, (x, y))$.

**Definition 1** (Proper Probabilistically Robust PAC Learnability). *Let $\ell_{\mathcal{G}, \mu}(h, (x, y))$ denote an arbitrary probabilistically robust loss function. For any $\epsilon, \delta \in (0, 1)$, the sample complexity of probabilistically robust $(\epsilon, \delta)$-learning $\mathcal{H}$ with respect to $\ell_{\mathcal{G}, \mu}$, denoted $n(\epsilon, \delta; \mathcal{H}, \ell_{\mathcal{G}, \mu})$ is the smallest number $m \in \mathbb{N}$ for which there exists a* proper *learning rule $\mathcal{A} : (\mathcal{X} \times \mathcal{Y})^* \to \mathcal{H}$ such that for every distribution $\mathcal{D}$ over $\mathcal{X} \times \mathcal{Y}$, with probability at least $1 - \delta$ over $S \sim \mathcal{D}^m$,*

$$\mathbb{E}_{\mathcal{D}}[\ell_{\mathcal{G}, \mu}(\mathcal{A}(S), (x, y))] \leq \inf_{h \in \mathcal{H}} \mathbb{E}_{\mathcal{D}}[\ell_{\mathcal{G}, \mu}(h, (x, y))] + \epsilon.$$

*We say that $\mathcal{H}$ is proper probabilistically robustly PAC learnable with respect to $\ell_{\mathcal{G}, \mu}$ if $\forall \epsilon, \delta \in (0, 1)$, $n(\epsilon, \delta; \mathcal{H}, \ell_{\mathcal{G}, \mu})$ is finite.*

An important class of proper learning rules is ERMs which simply output the hypothesis $h \in \mathcal{H}$ that minimizes the loss $\ell_{\mathcal{G}, \mu}$ over the training sample. In this paper, we ultimately show that for a wide family of probabilistically robust loss functions, ERM is a proper learner according to Definition 1. On the other hand, in Section 3, we show that proper probabilistically robust PAC learning is not always possible for the loss function $\ell_{\mathcal{G}, \mu}^{\rho}(h, (x, y))$.

## 2.3 Complexity Measures

Under the standard 0-1 risk, the Vapnik-Chervonenkis dimension (VC dimension) plays an important role in characterizing PAC learnability, and more specifically, when ERM is possible. A hypothesis class $\mathcal{H}$ is PAC learnable with respect to the 0-1 loss if and only if its VC dimension is finite [Vapnik and Chervonenkis, 1971].

**Definition 2** (VC dimension). *A set $\{x_1, ..., x_n\} \in \mathcal{X}$ is shattered by $\mathcal{H}$, if $\forall y_1, ..., y_n \in \mathcal{Y}$, $\exists h \in \mathcal{H}$, such that $\forall i \in [n]$, $h(x_i) = y_i$. The VC dimension of $\mathcal{H}$, denoted $\mathrm{VC}(\mathcal{H})$, is defined as the largest natural number $n \in \mathbb{N}$ such that there exists a set $\{x_1, ..., x_n\} \in \mathcal{X}$ that is shattered by $\mathcal{H}$.*

One *sufficient* condition for proper learning, based on Vapnik's "General Learning" [Vapnik, 2006], is the finiteness of the VC dimension of a binary loss class

$$\mathcal{L}^{\mathcal{H}} := \{(x, y) \mapsto \ell(h, (x, y)) : h \in \mathcal{H}\}$$

where $\ell(h, (x, y))$ is some loss function mapping to $\{0, 1\}$. In particular, if the VC dimension of the loss class $\mathcal{L}^{\mathcal{H}}$ is finite, then $\mathcal{H}$ is PAC learnable with respect to $\ell$ using ERM with sample complexity that scales linearly with $\mathrm{VC}(\mathcal{L}^{\mathcal{H}})$. In this sense, if one can upper bound $\mathrm{VC}(\mathcal{L}^{\mathcal{H}})$ in terms of $\mathrm{VC}(\mathcal{H})$, then finite VC dimension is sufficient for proper learnability. Unfortunately, for adversarially robust learning, Montasser et al. [2019] show that there can be an arbitrary gap between the VC dimension of the adversarially robust loss class $\mathcal{L}_{\mathcal{G}}^{\mathcal{H}} := \{(x, y) \mapsto \ell_{\mathcal{G}}(h, (x, y)) : h \in \mathcal{H}\}$ and the VC dimension of $\mathcal{H}$. Likewise, in Section 3, we show that for the $\rho$-probabilistically robust loss $\ell_{\mathcal{G}, \mu}^{\rho}$, there can also be an arbitrarily large gap between the VC dimension of the loss class and the VC dimension of the hypothesis class.

As many of the loss functions we consider will actually map to values in $\mathbb{R}$, the VC dimension of the loss class will not be well-defined. Instead, we can capture the complexity of the loss class via the *empirical* Rademacher complexity.

**Definition 3** (Empirical Rademacher Complexity of Loss Class). *Let $\ell$ be a loss function, $S = \{(x_1, y_1), ..., (x_n, y_n)\} \in (\mathcal{X} \times \mathcal{Y})^*$ be any sequence of examples, and $\mathcal{L}^{\mathcal{H}} = \{(x, y) \mapsto \ell(h, (x, y)) : h \in \mathcal{H}\}$ be a loss class. The empirical Rademacher complexity of $\mathcal{L}^{\mathcal{H}}$ is defined as*

$$\hat{\mathfrak{R}}_m(\mathcal{L}^{\mathcal{H}}) = \mathbb{E}_\sigma \left[ \sup_{f \in \mathcal{L}^{\mathcal{H}}} \left( \frac{1}{n} \sum_{i=1}^{m} \sigma_i f(x_i, y_i) \right) \right]$$

*where $\sigma_1, ..., \sigma_m$ are independent* Rademacher *random variables.*

A standard result relates the empirical Rademacher complexity to the generalization error of hypotheses in $\mathcal{H}$ with respect to a real-valued bounded loss function $\ell(h, (x, y))$ [Bartlett and Mendelson, 2002].

**Theorem 2.1** (Rademacher-based Uniform Convergence). *Let $\mathcal{D}$ be a distribution over $\mathcal{X} \times \mathcal{Y}$ and $\ell(h, (x, y)) \leq c$ be a bounded loss function. With probability at least $1 - \delta$ over the sample $S \sim \mathcal{D}^m$, for all $h \in \mathcal{H}$ simultaneously,*

$$\left| \mathbb{E}_\mathcal{D}[\ell(h(x), y)] - \hat{\mathbb{E}}_S[\ell(h(x), y)] \right| \leq 2\hat{\mathfrak{R}}_m(\mathcal{F}) + O\left( c\sqrt{\frac{\ln(\frac{1}{\delta})}{n}} \right)$$

*where $\hat{\mathbb{E}}_S[\ell(h(x), y)] = \frac{1}{|S|} \sum_{(x, y) \in S} \ell(h(x), y)$ is the empirical average of the loss over $S$.*

## 3 Not All Robust Loss Relaxations Enable Proper Learning

Recall the $\rho$-probabilistically robust loss

$$\ell^{\rho}_{\mathcal{G},\mu}(h, (x, y)) := \mathbb{1}\{\mathbb{P}_{g \sim \mu}(h(g(x)) \neq y) > \rho\},$$

and its corresponding risk $R^{\rho}_{\mathcal{G},\mu}(h; \mathcal{D}) := \mathbb{E}_{(x,y) \sim \mathcal{D}}\left[\ell^{\rho}_{\mathcal{G},\mu}(h, (x, y))\right]$. At a high-level, proper learning under the $\ell^{\rho}_{\mathcal{G},\mu}$ requires finding a hypothesis $h \in \mathcal{H}$ that is robust to at least a $1 - \rho$ fraction of the perturbations in $\mathcal{G}$ for each example in the support of the data distribution $\mathcal{D}$.

Which hypothesis classes are properly learnable with respect to $\ell^{\rho}_{\mathcal{G},\mu}$ according to Definition 1? In Appendix B.1, we show that if $\mathcal{G}$ is finite, then finite VC dimension is sufficient for proper learning with respect to $\ell^{\rho}_{\mathcal{G},\mu}$. On the other hand, in this section, we show that if $\mathcal{G}$ is allowed to be arbitrary, VC dimension is not sufficient for *proper* learning with respect to $\ell^{\rho}_{\mathcal{G},\mu}$, let alone learning via ERM.

**Theorem 3.1.** *There exists $(\mathcal{G}, \mu)$ such that for every $\rho \in [0, 1)$, there exists a hypothesis class $\mathcal{H} \subseteq \mathcal{Y}^{\mathcal{X}}$ with $\mathrm{VC}(\mathcal{H}) \leq 1$ such that $\mathcal{H}$ is not properly probabilistically robust PAC learnable with respect to $\ell^{\rho}_{\mathcal{G},\mu}$.*

To prove Theorem 3.1, we fix $\mathcal{X} = \mathbb{R}^d$, $\mathcal{G} = \{g_\delta : \delta \in \mathbb{R}^d, ||\delta||_p \leq \gamma\}$ such that $g_\delta(x) = x + \delta$ for all $x \in \mathcal{X}$ for some $\gamma > 0$, and $\mu$ to be the uniform measure over $\mathcal{G}$. In other words, we are picking our perturbation sets to be $\ell_p$ balls of radius $\gamma$ and our perturbation measures to be uniform over each perturbation set. Note that by construction of $\mathcal{G}$, a uniform measure $\mu$ over $\mathcal{G}$ also induces a uniform measure $\mu_x$ over $\mathcal{G}(x) := \{g_\delta(x) : g_\delta \in \mathcal{G}\} \subset \mathbb{R}^d$. We start by showing that for every $\rho \in [0, 1)$, there can be an arbitrary gap between the VC dimension of $\mathcal{H}$ and the loss class $\mathcal{L}^{\mathcal{H},\rho}_{\mathcal{G},\mu} := \{(x, y) \mapsto \ell^{\rho}_{\mathcal{G},\mu}(h, (x, y)) : h \in \mathcal{H}\}$.

**Lemma 3.2.** *For every $\rho \in [0, 1)$ and $m \in \mathbb{N}$, there exists a hypothesis class $\mathcal{H} \subseteq \mathcal{Y}^{\mathcal{X}}$ such that $\mathrm{VC}(\mathcal{H}) \leq 1$ but $\mathrm{VC}(\mathcal{L}^{\mathcal{H},\rho}_{\mathcal{G},\mu}) \geq m$.*

The proof of Lemma 3.2 is found in Appendix B.2. We highlight two key differences between Lemma 3.2 and its analog, Lemma 2, in Montasser et al. [2019]. First, we need to provide *both* a perturbation set and a perturbation measure. The interplay between these two objects is not present in Montasser et al. [2019] and, apriori, it is not clear that these would indeed be $\ell_p$ balls and the uniform measure. Second, in order for a hypothesis to be probabilistically non-robust there needs to exist a large enough *region* of perturbations over which it makes mistakes. This is in contrast to Montasser et al. [2019], where a hypothesis is adversarially non-robust as long as there exists *one* non-robust perturbation. Constructing a hypothesis class that achieves all possible probabilistically robust loss behaviors while also having low VC dimension is non-trivial - we need hypotheses to be expressive enough to have large regions of non-robustness while not being too expressive such that VC dimension increases.

Next, we show that the hypothesis class construction in Lemma 3.2 can be used to show the existence of a hypothesis class that cannot be learned properly. Lemma 3.3 is similar to Lemma 3 in Montasser et al. [2019] and is proved in Appendix B.3.

**Lemma 3.3.** *For every $\rho \in [0, 1)$ and $m \in \mathbb{N}$ there exists $\mathcal{H} \subseteq \mathcal{Y}^{\mathcal{X}}$ with $\mathrm{VC}(\mathcal{H}) \leq 1$ such that for any proper learner $\mathcal{A} : (\mathcal{X} \times \mathcal{Y})^* \to \mathcal{H}$: (1) there is a distribution $\mathcal{D}$ over $\mathcal{X} \times \mathcal{Y}$ and a hypothesis $h^* \in \mathcal{H}$ where $R^{\rho}_{\mathcal{G},\mu}(h^*; \mathcal{D}) = 0$ and (2) with probability at least $1/7$ over $S \sim D^m$, $R^{\rho}_{\mathcal{G},\mu}(\mathcal{A}(S); \mathcal{D}) > 1/8$.*

Finally, the proof of Theorem 3.1 uses Lemma 3.3 and follows a similar idea as its analog in Montasser et al. [2019] (Theorem 1). However, since our hypothesis class construction in Lemma 3.2 is different, some subtle modifications need to be made. We include a complete proof in Appendix B.4.

## 4 Proper Learnability Under Relaxed Losses

Despite the fact that VC classes are not $\rho$-probabilistically robust learnable using proper learning rules, our framework still enables us to capture a wide range of relaxations to the adversarially robust loss for which proper learning is possible.

In particular, consider robust loss relaxations of the form $\ell_{\mathcal{G},\mu}(h, (x, y)) = \ell(y \mathbb{E}_{g \sim \mu}[h(g(x))])$ where $\ell(t) : \mathbb{R} \to \mathbb{R}$ is a $L$-Lipschitz function. This class of loss functions is general, capturing many natural robust loss relaxations like the hinge loss $1 - y \mathbb{E}_{g \sim \mu}[h(g(x))]$, squared loss $(y -$

$\mathbb{E}_{g\sim\mu}\left[h(g(x))\right])^2 = (1 - y\mathbb{E}_{g\sim\mu}\left[h(g(x))\right])^2$, and exponential loss $e^{-y\mathbb{E}_{g\sim\mu}[h(g(x))]}$. Furthermore, the class of Lipschitz functions $\ell : \mathbb{R} \to \mathbb{R}$ on the margin $y\mathbb{E}_{g\sim\mu}\left[h(g(x))\right]$ enables us to capture levels of robustness between the average- and worst-case. For example, taking $\ell(t) = \frac{1-t}{2}$ results in the loss $\ell_{\mathcal{G},\mu}(h, (x, y)) = \ell(y\mathbb{E}_{g\sim\mu}\left[h(g(x))\right]) = \mathbb{P}_{g\sim\mu}\left[h(g(x)) \neq y\right]$, corresponding to average-case robustness, or *data augmentation*. On the other hand, taking $\ell(t) = \min(\frac{1-t}{2\rho}, 1)$ for some $\rho \in (0, 1)$, results in the loss

$$\ell_{\mathcal{G},\mu}(h, (x, y)) = \ell(y\mathbb{E}_{g\sim\mu}\left[h(g(x))\right]) = \min\left(\frac{\mathbb{P}_{g\sim\mu}\left[h(g(x)) \neq y\right]}{\rho}, 1\right)$$

which corresponds to a notion of robustness that becomes stricter as $\rho$ approaches $0$. We note that some of the losses in our family were studied by Rice et al. [2021]. However, their focus was on evaluating robustness, while ours is about (proper) learnability.

Lemma 4.1 shows that for hypothesis classes $\mathcal{H}$ with finite VC dimension, for any $(\mathcal{G}, \mu)$, all $L$-Lipschitz loss functions $\ell_{\mathcal{G},\mu}(h, (x, y))$ enjoy the uniform convergence property.

**Lemma 4.1** (Uniform Convergence of Lipschitz Loss). *Let $\mathcal{H}$ be a hypothesis class with finite VC dimension, $(\mathcal{G}, \mu)$ be a perturbation set and measure, and $\ell_{\mathcal{G},\mu}(h, (x, y)) = \ell(y\mathbb{E}_{g\sim\mu}\left[h(g(x))\right])$ such that $\ell : \mathbb{R} \to \mathbb{R}$ is a $L$-Lipschitz function. With probability at least $1 - \delta$ over a sample $S \sim \mathcal{D}^n$ of size $n = O\left(\frac{\mathrm{VC}(\mathcal{H})L^2 \ln(\frac{L}{\epsilon}) + \ln(\frac{1}{\delta})}{\epsilon^2}\right)$, for all $h \in \mathcal{H}$ simultaneously,*

$$\left|\mathbb{E}_{\mathcal{D}}\left[\ell_{\mathcal{G},\mu}(h, (x, y))\right] - \hat{\mathbb{E}}_S\left[\ell_{\mathcal{G},\mu}(h, (x, y))\right]\right| \leq \epsilon.$$

*Proof.* Let $\mathrm{VC}(\mathcal{H}) = d$ and $S = \{(x_1, y_1), ..., (x_m, y_m)\}$ be a set of examples drawn i.i.d from $\mathcal{D}$. Define the loss class $\mathcal{L}_{\mathcal{G},\mu}^{\mathcal{H}} = \{(x, y) \mapsto \ell_{\mathcal{G},\mu}(h, (x, y)) : h \in \mathcal{H}\}$. Observe that we can reparameterize $\mathcal{L}_{\mathcal{G},\mu}^{\mathcal{H}}$ as the composition of a $L$-Lipschitz function $\ell(x)$ and the function class $\mathcal{F}_{\mathcal{G},\mu}^{\mathcal{H}} = \{(x, y) \mapsto y\mathbb{E}_{g\sim\mu}\left[h(g(x))\right] : h \in \mathcal{H}\}$. By Proposition 2.1, to show the uniform convergence property of $\ell_{\mathcal{G},\mu}(h, (x, y))$, it suffices to upper bound $\hat{\mathfrak{R}}_m(\mathcal{L}_{\mathcal{G},\mu}^{\mathcal{H}}) = \hat{\mathfrak{R}}_m(\ell \circ \mathcal{F}_{\mathcal{G},\mu}^{\mathcal{H}})$, the empirical Rademacher complexity of the loss class. Since $\ell$ is $L$-Lipschitz, by Ledoux-Talagrand's contraction principle [Ledoux and Talagrand, 1991], it follows that $\hat{\mathfrak{R}}_m(\mathcal{L}_{\mathcal{G},\mu}^{\mathcal{H}}) = \hat{\mathfrak{R}}_m(\ell \circ \mathcal{F}_{\mathcal{G},\mu}^{\mathcal{H}}) \leq L \cdot \hat{\mathfrak{R}}_m(\mathcal{F}_{\mathcal{G},\mu}^{\mathcal{H}})$. Thus, it actually suffices to upperbound $\hat{\mathfrak{R}}_m(\mathcal{F}_{\mathcal{G},\mu}^{\mathcal{H}})$ instead. Starting with the definition of the empirical Rademacher complexity:

$$\hat{\mathfrak{R}}_m(\mathcal{F}_{\mathcal{G},\mu}^{\mathcal{H}}) = \frac{1}{m}\mathbb{E}_{\sigma\sim\{\pm1\}^m}\left[\sup_{h\in\mathcal{H}}\left(\sum_{i=1}^m \sigma_i y_i \mathbb{E}_{g\sim\mu}\left[h(g(x_i))\right]\right)\right]$$

$$= \frac{1}{m}\mathbb{E}_{\sigma\sim\{\pm1\}^m}\left[\sup_{h\in\mathcal{H}}\left(\mathbb{E}_{g\sim\mu}\left[\sum_{i=1}^m \sigma_i h(g(x_i))\right]\right)\right]$$

$$\leq \mathbb{E}_{g\sim\mu}\left[\frac{1}{m}\mathbb{E}_{\sigma\sim\{\pm1\}^m}\left[\sup_{h\in\mathcal{H}}\sum_{i=1}^m \sigma_i h(g(x_i))\right]\right],$$

where the last inequality follows from Jensen's inequality and Fubini's Theorem. Note that the quantity $\frac{1}{m}\mathbb{E}_{\sigma\sim\{\pm1\}^m}\left[\sup_{h\in\mathcal{H}}\sum_{i=1}^m \sigma_i h(g(x_i))\right]$ is the empirical Rademacher complexity of the hypothesis class $\mathcal{H}$ over the sample $\{g(x_1), ..., g(x_m)\}$ drawn i.i.d from the distribution defined by first sampling from the marginal data distribution, $x \sim \mathcal{D}_{\mathcal{X}}$, and then applying the transformation $g(x)$. By standard VC arguments, $\hat{\mathfrak{R}}_m(\mathcal{H}) \leq O\left(\sqrt{\frac{d \ln(\frac{m}{d})}{m}}\right)$, which implies that $\hat{\mathfrak{R}}_m(\mathcal{F}_{\mathcal{G},\mu}^{\mathcal{H}}) \leq \mathbb{E}_{g\sim\mu}\left[\hat{\mathfrak{R}}_m(\mathcal{H})\right] \leq O\left(\sqrt{\frac{d \ln(\frac{m}{d})}{m}}\right)$. Putting things together, we get $\hat{\mathfrak{R}}_m(\mathcal{L}_{\mathcal{G},\mu}^{\mathcal{H}}) = \hat{\mathfrak{R}}_m(\ell \circ \mathcal{F}_{\mathcal{G},\mu}^{\mathcal{H}}) \leq O\left(\sqrt{\frac{dL^2 \ln(\frac{m}{d})}{m}}\right)$. Proposition 2.1 then implies that with probability $1 - \delta$ over a sample $S \sim \mathcal{D}^m$ of size $m = O\left(\frac{dL^2 \ln(\frac{L}{\epsilon}) + \ln(\frac{1}{\delta})}{\epsilon^2}\right)$, we have

$$\left|\mathbb{E}_{\mathcal{D}}\left[\ell_{\mathcal{G},\mu}(h, (x, y))\right] - \hat{\mathbb{E}}_S\left[\ell_{\mathcal{G},\mu}(h, (x, y))\right]\right| \leq \epsilon$$

for all $h \in \mathcal{H}$ simultaneously. $\qquad\square$

Uniform convergence of Lipschitz-losses immediately implies proper learning via ERM.

**Theorem 4.2.** *Let* $\ell_{\mathcal{G},\mu}(h,(x,y)) = \ell(y\mathbb{E}_{g \sim \mu}[h(g(x))])$ *such that* $\ell : \mathbb{R} \to \mathbb{R}$ *is a L-Lipschitz function. For every hypothesis class* $\mathcal{H}$*, perturbation set and measure* $(\mathcal{G},\mu)$*, and* $(\epsilon,\delta) \in (0,1)^2$*, the proper learning rule* $\mathcal{A}(S) = \arg\min_{h \in \mathcal{H}} \hat{\mathbb{E}}_S[\ell_{\mathcal{G},\mu}(h,(x,y))]$*, for any distribution* $\mathcal{D}$ *over* $\mathcal{X} \times \mathcal{Y}$*, achieves, with probability at least* $1 - \delta$ *over a sample* $S \sim \mathcal{D}^n$ *of size* $n \geq O\left(\frac{\mathrm{VC}(\mathcal{H})L^2 \ln(\frac{L}{\epsilon}) + \ln(\frac{1}{\delta})}{\epsilon^2}\right)$*, the guarantee*

$$\mathbb{E}_{\mathcal{D}}[\ell_{\mathcal{G},\mu}(\mathcal{A}(S),(x,y))] \leq \inf_{h \in \mathcal{H}} \mathbb{E}_{\mathcal{D}}[\ell_{\mathcal{G},\mu}(h,(x,y))] + \epsilon.$$

At a high-level, Theorem 4.2 shows finite VC dimension is sufficient for achieving robustness *between* the average- and worst-case using ERM. In fact, the next theorem, whose proof can be found in Appendix C.2, shows that finite VC dimension may not even be necessary for this to be true. The proof of Theorem 4.3 involves considering the well-known infinite VC class $\mathcal{H} = \{x \mapsto \mathrm{sign}(\sin(wx)) : w \in \mathbb{R}\}$ and picking $(\mathcal{G},\mu)$ such that $\mathbb{E}_{g \sim \mu}[h(g(x))]$ is essentially constant in $x$ for all hypothesis $h \in \mathcal{H}$.

**Theorem 4.3.** *Let* $\ell_{\mathcal{G},\mu}(h,(x,y)) = \ell(y\mathbb{E}_{g \sim \mu}[h(g(x))])$ *such that* $\ell : \mathbb{R} \to \mathbb{R}$ *is a L-Lipschitz function. There exists* $\mathcal{H}$ *and* $(\mathcal{G},\mu)$ *such that* $\mathrm{VC}(\mathcal{H}) = \infty$ *but* $\mathcal{H}$ *is still properly learnable with respect to* $\ell_{\mathcal{G},\mu}(h,(x,y))$*.*

Together, Theorems 4.2 and 4.3 showcase an interesting trade-off. Theorem 4.3 indicates that by carefully choosing $(\mathcal{G},\mu)$, the complexity of $\mathcal{H}$ can be essentially smoothed out. On the other hand, Theorem 4.2 shows that any complexity in $(\mathcal{G},\mu)$ can be smoothed out if $\mathcal{H}$ has finite VC dimension. This interplay between the complexities of $\mathcal{H}$ and $(\mathcal{G},\mu)$ closely matches the intuition of Chapelle et al. [2000] in their work on Vicinal Risk Minimization. Note that the results in this section do not contradict that of Section 3 because $\ell_{\mathcal{G},\mu}^\rho(h,(x,y))$ is a *non-Lipschitz* function of $y\mathbb{E}_{g \sim \mu}[h(g(x))]$.

We end this section by noting that Lipschitzness of $\ell_{\mathcal{G},\mu}$ is sufficient but, in full generality, not necessary for proper learnability. For example, the loss function that completely ignores $(\mathcal{G},\mu)$ and just computes the 0-1 loss is not Lipschitz, however, it is learnable via ERM when the VC dimension of $\mathcal{H}$ is finite.

## 5 Proper Learnability Under Relaxed Competition

The results of Section 3 show that relaxing the adversarially robust loss may not always enable proper learning, even for very natural robust loss relaxations. In this section, we show that this bottleneck can be alleviated if we also allow the learner to compete against a slightly stronger notion of robustness. Furthermore, we expand on this idea by exploring other robust learning settings where allowing the learner to compete against a stronger notion of robustness enables proper learnability. We denote this type of modification to the standard adversarially and probabilistically robust leaning settings as robust learning under *relaxed competition*. Prior works on Tolerantly Robust PAC Learning [Ashtiani et al., 2022, Bhattacharjee et al., 2022] mentioned in the introduction fit under this umbrella.

Our main tool in this section is Lemma 5.1, which we term as Sandwich Uniform Convergence (SUC). Roughly speaking, SUC provides a sufficient condition under which ERM outputs a predictor that generalizes well with respect to a stricter notion of loss. A special case of SUC has implicitly appeared in margin theory (e.g., see Mohri et al. [2018, Section 5.4]), where one evaluates the 0-1 risk of the output hypothesis against the optimal *margin* 0-1 risk.

**Lemma 5.1** (Sandwich Uniform Convergence)**.** *Let* $\ell_1(h,(x,y))$ *and* $\ell_2(h,(x,y))$ *be bounded, non-negative loss functions such that for all* $h \in \mathcal{H}$ *and* $(x,y) \in \mathcal{X} \times \mathcal{Y}$*, we have* $\ell_1(h,(x,y)) \leq \ell_2(h,(x,y)) \leq 1$*. If there exists a loss function* $\tilde{\ell}(h,(x,y))$ *such that* $\ell_1(h,(x,y)) \leq \tilde{\ell}(h,(x,y)) \leq \ell_2(h,(x,y))$ *and* $\tilde{\ell}(h,(x,y))$ *enjoys the* uniform convergence *property with sample complexity* $n(\epsilon,\delta)$*, then the learning rule* $\mathcal{A}(S) = \inf_{h \in \mathcal{H}} \hat{\mathbb{E}}_S[\ell_2(h,(x,y))]$ *achieves, with probability* $1 - \delta$ *over a sample* $S \sim \mathcal{D}^m$ *of size* $m \geq n(\epsilon/2,\delta/2) + O\left(\frac{\ln(\frac{1}{\delta})}{\epsilon^2}\right)$*, the guarantee*

$$\mathbb{E}_{\mathcal{D}}[\ell_1(\mathcal{A}(S),(x,y))] \leq \inf_{h \in \mathcal{H}} \mathbb{E}_{\mathcal{D}}[\ell_2(h,(x,y))] + \epsilon.$$

*Proof.* Let $\mathcal{A}(S) = \inf_{h \in \mathcal{H}} \mathbb{E}_S [\ell_2(h, (x, y))]$. By uniform convergence of $\tilde{\ell}(h, (x, y))$, we have that for sample size $m = n(\frac{\epsilon}{2}, \frac{\delta}{2})$, with probability at least $1 - \frac{\delta}{2}$, over a sample $S \sim \mathcal{D}^m$, for every hypothesis $h \in \mathcal{H}$ simultaneously,

$$\mathbb{E}_\mathcal{D}\left[\tilde{\ell}(h, (x, y))\right] \le \hat{\mathbb{E}}_S\left[\tilde{\ell}(h, (x, y))\right] + \frac{\epsilon}{2}.$$

In particular, this implies that for $\hat{h} = \mathcal{A}(S)$, we have

$$\mathbb{E}_\mathcal{D}\left[\tilde{\ell}(\hat{h}, (x, y))\right] \le \hat{\mathbb{E}}_S\left[\tilde{\ell}(\hat{h}, (x, y))\right] + \frac{\epsilon}{2}.$$

Since, $\ell_1(h, (x, y)) \le \tilde{\ell}(h, (x, y)) \le \ell_2(h, (x, y))$, we have that

$$\mathbb{E}_\mathcal{D}\left[\ell_1(\hat{h}, (x, y))\right] \le \hat{\mathbb{E}}_S[\ell_2(h^*, (x, y))] + \frac{\epsilon}{2}$$

where $h^* = \inf_{h \in \mathcal{H}} \mathbb{E}_\mathcal{D}[\ell_2(h, (x, y))]$. It now remains to upper bound $\hat{\mathbb{E}}_S[\ell_2(h^*, (x, y))]$ with high probability. However, a standard Hoeffding bound tells us that with probability $1 - \frac{\delta}{2}$ over a sample $S$ of size $O(\frac{\ln(\frac{1}{\delta})}{\epsilon^2})$, $\hat{\mathbb{E}}_S[\ell_2(h^*, (x, y))] \le \mathbb{E}_\mathcal{D}[\ell_2(h^*, (x, y))] + \frac{\epsilon}{2}$. Thus, by union bound, we get that with probability at least $1 - \delta$, $\mathbb{E}_\mathcal{D}\left[\ell_1(\hat{h}, (x, y))\right] \le \mathbb{E}_\mathcal{D}[\ell_2(h^*, (x, y))] + \epsilon$, using a sample of size $n(\epsilon/2, \delta/2) + O(\frac{\ln(\frac{1}{\delta})}{\epsilon^2})$. $\qquad\square$

Lemma 5.1 only requires the *existence* of such a sandwiched loss function that enjoys uniform convergence—we do not actually require it to be computable. In the next two sections, we exploit this fact to give three new generalization guarantees for the empirical risk minimizer over the adversarially robust loss $\ell_\mathcal{G}(h, (x, y))$ and $\rho$-probabilistically robust loss $\ell_{\mathcal{G}, \mu}^\rho(h, (x, y))$, hereafter denoted by $\text{RERM}(S; \mathcal{G}) := \arg\min_{h \in \mathcal{H}} \hat{\mathbb{E}}_S[\ell_\mathcal{G}(h, (x, y))]$ and $\text{PRERM}(S; (\mathcal{G}, \mu), \rho) := \arg\min_{h \in \mathcal{H}} \hat{\mathbb{E}}_S\left[\ell_{\mathcal{G}, \mu}^\rho(h, (x, y))\right]$ respectively.

## 5.1 $(\rho, \rho^*)$-Probabilistically Robust PAC Learning

In light of the hardness result of Section 3, we slightly tweak the learning setup in Definition 1 by allowing $\mathcal{A}$ to compete against the hypothesis minimizing the probabilistic robust risk at a level $\rho^* < \rho$. Under this further relaxation, we show that proper learning becomes possible, and that too, via PRERM. In particular, Theorem 5.2 shows that while VC classes are not properly $\rho$-probabilistically robust PAC learnable, they are properly $(\rho, \rho^*)$-probabilistically robust PAC learnable.

**Theorem 5.2** (Proper $(\rho, \rho^*)$-Probabilistically Robust PAC Learner). *Let $0 \le \rho^* < \rho$. Then, for every hypothesis class $\mathcal{H}$, perturbation set and measure $(\mathcal{G}, \mu)$, and $(\epsilon, \delta) \in (0, 1)^2$, the proper learning rule $\mathcal{A}(S) = \text{PRERM}(S; (\mathcal{G}, \mu), \rho^*)$, for any distribution $\mathcal{D}$ over $\mathcal{X} \times \mathcal{Y}$, achieves, with probability at least $1 - \delta$ over a sample $S \sim \mathcal{D}^n$ of size $n \ge O\left(\frac{\frac{\text{VC}(\mathcal{H})}{(\rho - \rho^*)^2} \ln(\frac{1}{(\rho - \rho^*)\epsilon}) + \ln(\frac{1}{\delta})}{\epsilon^2}\right)$, the guarantee*

$$R_{\mathcal{G}, \mu}^\rho(\mathcal{A}(S); \mathcal{D}) \le \inf_{h \in \mathcal{H}} R_{\mathcal{G}, \mu}^{\rho^*}(h; \mathcal{D}) + \epsilon.$$

In contrast to Section 3, where proper learning is not always possible, Theorem 5.2 shows that if we compare our learner to the best hypothesis for a *slightly* stronger level of probabilistic robustness, then not only is proper learning possible for VC classes, but it is possible via ERM. Our main technique to prove Theorem 5.2 is to consider a *different* probabilistically robust loss function that is (1) a Lipschitz function of $y\mathbb{E}_{g \sim \mu}[h(g(x)) \ne y]$ and (2) can be sandwiched in between $\ell_{\mathcal{G}, \mu}^{\rho^*}$ and $\ell_{\mathcal{G}, \mu}^\rho$. Then, Theorem 5.2 follows from Lemma 5.1. The full proof is in Appendix D.1.

## 5.2 $(\rho, \mathcal{G})$-Probabilistically Robust PAC Learning

Can measure-independent learning guarantees be achieved if we instead compare the learner's probabilistically robust risk $R_{\mathcal{G}, \mu}^\rho$ to the best *adversarially robust risk* $R_\mathcal{G}$ over $\mathcal{H}$? We answer this in the affirmative by using SUC. We show that if one wants to compete against the best hypothesis for the worst-case adversarially robust risk, it is sufficient to run RERM.

**Theorem 5.3** (Proper $(\rho, \mathcal{G})$-Probabilistically Robust PAC Learner)**.** *For every hypothesis class $\mathcal{H}$, perturbation set $\mathcal{G}$, and $(\epsilon, \delta) \in (0,1)^2$, the proper learning rule $\mathcal{A}(S) = \text{RERM}(S; \mathcal{G})$, for any measure $\mu$ over $\mathcal{G}$ and any distribution $\mathcal{D}$ over $\mathcal{X} \times \mathcal{Y}$, achieves, with probability at least $1 - \delta$ over a sample $S \sim \mathcal{D}^n$ of size $n \geq O\left(\frac{\frac{\text{VC}(\mathcal{H})}{\rho^2}\ln(\frac{1}{\rho\epsilon}) + \ln(\frac{1}{\delta})}{\epsilon^2}\right)$, the guarantee*

$$R^\rho_{\mathcal{G},\mu}(\mathcal{A}(S); \mathcal{D}) \leq \inf_{h \in \mathcal{H}} R_{\mathcal{G}}(h; \mathcal{D}) + \epsilon.$$

The proof of Theorem 5.3 can be found in Appendix D.2, which follows directly from Lemma 5.1 by a suitable choice of the sandwiched loss $\ell$. We make a few remarks about the practical importance of Theorem 5.3. Theorem 5.3 implies that for any pre-specified perturbation function class $\mathcal{G}$ (for example $\ell_p$ balls), running RERM is sufficient to obtain a hypothesis that is probabilistically robust with respect to *any* fixed measure $\mu$ over $\mathcal{G}$. Moreover, the level of robustness of the predictor output by RERM, as measured by $1 - \rho$, scales directly with the sample size - the more samples one has, the smaller $\rho$ can be made. Alternatively, for a fixed sample size $m$, desired error $\epsilon$ and confidence $\delta$, one can use the sample complexity guarantee in Theorem 5.3 to back-solve the robustness guarantee $\rho$.

We highlight that the generalization bound in Theorem 5.3 is a *measure-independent* guarantee. This means that $\rho$ does not quantify the level of robustness of the output hypothesis with respect to any one particular measure, but for *any* measure. This is desirable, as in contrast to the previous section, the $\rho$ here more succinctly quantifies the level of robustness achieved by the output classifier. Lastly, we highlight that while the sample complexity of adversarially robust PAC learning can be exponential in the VC dimension of $\mathcal{H}$ [Montasser et al., 2019], this is not the case for $(\rho, \mathcal{G})$-probabilistically robust PAC learning, where we only get a linear dependence on VC dimension.

## 5.3 Tolerantly Robust PAC Learning

In Tolerantly Robust PAC Learning [Bhattacharjee et al., 2022, Ashtiani et al., 2022], the learner's adversarially robust risk under a perturbation set $\mathcal{G}$ is compared with the best achievable adversarially robust risk for a larger perturbation set $\mathcal{G}' \supset \mathcal{G}$. Ashtiani et al. [2022] study the setting where both $\mathcal{G}$ and $\mathcal{G}'$ induce $\ell_p$ balls with radius $r$ and $(1 + \gamma)r$ respectively. In the work of Bhattacharjee et al. [2022], $\mathcal{G}$ is arbitrary, but $\mathcal{G}'$ is constructed such that it induces perturbation sets that are the union of balls with radius $\gamma$ that cover $\mathcal{G}$. Critically, Bhattacharjee et al. [2022] show that, under certain assumptions, running RERM over a larger perturbation set $\mathcal{G}'$ is sufficient for Tolerantly Robust PAC learning. In this section, we take a slightly different approach to Tolerantly Robust PAC learning. Instead of having the learner compete against the best possible risk for a larger perturbation set, we have the learner still compete against the best possible adversarially robust risk over $\mathcal{G}$, but evaluate the learner's adversarially robust risk using a *smaller* perturbation set $\mathcal{G}' \subset \mathcal{G}$.

For what $\mathcal{G}' \subset \mathcal{G}$ is Tolerantly Robust PAC learning via RERM possible? As an immediate result of Lemma 5.1 and Vapnik's "General Learning", finite VC dimension of the loss class $\mathcal{L}^{\mathcal{H}}_{\mathcal{G}'} = \{(x,y) \mapsto \ell_{\mathcal{G}'}(h, (x,y)) : h \in \mathcal{H}\}$ is sufficient. Note that finite VC dimension of $\mathcal{L}^{\mathcal{H}}_{\mathcal{G}'}$ implies that the loss function $\ell_{\mathcal{G}'}(h, (x,y))$ enjoys the uniform convergence property with sample complexity $O\left(\frac{\text{VC}(\mathcal{L}^{\mathcal{G}'}_{\mathcal{H}}) + \ln(\frac{1}{\delta})}{\epsilon^2}\right)$. Thus, taking $\ell_1(h, (x,y)) = \tilde{\ell}(h, (x,y)) = \ell_{\mathcal{G}'}(h, (x,y))$ and $\ell_2(h, (x,y)) = \ell_{\mathcal{G}}(h, (x,y))$ in Lemma 5.1, we have that if there exists a $\mathcal{G}' \subset \mathcal{G}$ such that $\text{VC}(\mathcal{L}^{\mathcal{G}'}_{\mathcal{H}}) < \infty$, then with probability $1 - \delta$ over a sample $S \sim \mathcal{D}^n$ of size $n = O\left(\frac{\text{VC}(\mathcal{L}^{\mathcal{H}}_{\mathcal{G}'}) + \ln(\frac{1}{\delta})}{\epsilon^2}\right)$,

$$R_{\mathcal{G}'}(\mathcal{A}(S); \mathcal{D}) \leq \inf_{h \in \mathcal{H}} R_{\mathcal{G}}(h; \mathcal{D}) + \epsilon,$$

where $\mathcal{A}(S) = \text{RERM}(S; \mathcal{G})$.

Alternatively, if $\mathcal{G}' \subset \mathcal{G}$ such that there exists a *finite* subset $\tilde{\mathcal{G}} \subset \mathcal{G}$ where $\ell_{\mathcal{G}'}(h, (x,y)) \leq \ell_{\tilde{\mathcal{G}}}(h, (x,y))$, then Tolerantly Robust PAC learning via RERM is possible with sample complexity that scales according to $O\left(\frac{\text{VC}(\mathcal{H})\log(|\tilde{\mathcal{G}}|) + \ln(\frac{1}{\delta})}{\epsilon^2}\right)$. This result essentially comes from the fact that the VC dimension of the loss class for any finite perturbation set $\tilde{\mathcal{G}}$ incurs only a $\log(|\tilde{\mathcal{G}}|)$ blow-up from the VC dimension of $\mathcal{H}$ (see Lemma 1.1 in Attias et al. [2021]). Thus, finite VC dimension of

$\mathcal{H}$ implies finite VC dimension of the loss class $\mathcal{L}_{\mathcal{H}}^{\tilde{\mathcal{G}}}$ which implies uniform convergence of the loss $\ell_{\tilde{\mathcal{G}}}(h, (x, y))$, as needed for Lemma 5.1 to hold.

We now give an example where such a finite approximation of $\mathcal{G}'$ is possible. In order to do so, we will need to consider a *metric space* of perturbation functions $(\mathcal{G}, d)$ and define a notion of "nice" perturbation sets, similar to "regular" hypothesis classes from Bhattacharjee et al. [2022].

**Definition 4** (*r*-Nice Perturbation Set). *Let $\mathcal{H}$ be a hypothesis class and $(\mathcal{G}, d)$ a metric space of perturbation functions. Let $B_r(g) := \{g' \in \mathcal{G} : d(g, g') \leq r\}$ denote a closed ball of radius $r$ centered around $g \in \mathcal{G}$. We say that $\mathcal{G}' \subset \mathcal{G}$ is $r$-Nice with respect to $\mathcal{H}$, if for all $x \in \mathcal{X}$, $h \in \mathcal{H}$, and $g \in \mathcal{G}'$, there exists a $g^* \in \mathcal{G}$, such that $g \in B_r(g^*)$ and $h(g(x)) = h(g'(x))$ for all $g' \in B_r(g^*)$.*

Definition 4 prevents a situation where a hypothesis $h \in \mathcal{H}$ is non-robust to an isolated perturbation function $g \in \mathcal{G}'$ for any given labelled example $(x, y) \in \mathcal{X} \times \mathcal{Y}$. If a hypothesis $h$ is non-robust to a perturbation $g \in \mathcal{G}'$, then Definition 4 asserts that there must exist a small ball of perturbation functions in $\mathcal{G}$ over which $h$ is also non-robust. Next, we define the covering number.

**Definition 5** (Covering Number). *Let $(\mathcal{M}, d)$ be a metric space, let $\mathcal{K} \subset \mathcal{M}$ be a subset, and $r > 0$. Let $B_r(x) = \{x' \in \mathcal{M} : d(x, x') \leq r\}$ denote the ball of radius $r$ centered around $x \in \mathcal{M}$. A subset $\mathcal{C} \subset \mathcal{M}$ is an $r$-covering of $\mathcal{K}$ if $\mathcal{K} \subset \bigcup_{c \in \mathcal{C}} B_r(c)$. The covering number of $\mathcal{K}$, denoted $\mathcal{N}_r(\mathcal{K}, d)$, is the smallest cardinality of any $r$-covering of $\mathcal{K}$.*

Finally, let $\mathcal{G}'_{2r} = \bigcup_{g \in \mathcal{G}'} B_{2r}(g)$ denote the union over all balls of radius $2r$ with centers in $\mathcal{G}'$. Theorem 5.4 then states that if there exists a set $\mathcal{G}' \subset \mathcal{G}$ that is $r$-Nice with respect to $\mathcal{H}$, then Tolerantly Robust PAC learning is possible via RERM with sample complexity that scales logarithmically with $\mathcal{N}_r(\mathcal{G}'_{2r}, d)$.

**Theorem 5.4** (Tolerantly Robust PAC learning under Nice Perturbations). *Let $\mathcal{H} \subseteq \mathcal{Y}^{\mathcal{X}}$ be a hypothesis class and $(\mathcal{G}, d)$ be a metric space of perturbation functions. Given a subset $\mathcal{G}' \subset \mathcal{G}$ such that $\mathcal{G}'$ is $r$-Nice with respect to $\mathcal{H}$, then the proper learning rule $\mathcal{A}(S) = \text{RERM}(S; \mathcal{G})$, for any distribution $\mathcal{D}$ over $\mathcal{X} \times \mathcal{Y}$, achieves, with probability at least $1 - \delta$ over a sample $S \sim \mathcal{D}^n$ of size $n \geq O\left(\frac{\text{VC}(\mathcal{H}) \log(\mathcal{N}_r(\mathcal{G}'_{2r}, d)) + \ln(\frac{1}{\delta})}{\epsilon^2}\right)$, the guarantee*

$$R_{\mathcal{G}'}(\mathcal{A}(S); \mathcal{D}) \leq \inf_{h \in \mathcal{H}} R_{\mathcal{G}}(h; \mathcal{D}) + \epsilon.$$

In Appendix D, we give a full proof and show that $\ell_p$ *balls are $r$-Nice perturbation sets for robustly learning halfspaces*. Note that Theorem 5.4 does not require apriori knowledge of the $r$-Nice perturbation set $\mathcal{G}'$, but just *its existence*. Therefore, Theorem 5.4 applies to the largest possible $r$-Nice perturbation subset of $\mathcal{G}$. This is important from a practical standpoint as computing an $r$-Nice perturbation set might not be computationally tractable. Accordingly, while RERM may not be a proper adversarially robust PAC learner [Montasser et al., 2019], Theorem 5.4 shows that RERM is a proper tolerantly robust PAC learner.

## 6 Discussion

In this work, we show that there exists natural relaxations of the adversarially robust loss for which finite VC dimension is still not sufficient for proper learning. On the other hand, we identify a large set of Lipschitz robust loss relaxations for which finite VC dimension is sufficient for proper learning. In addition, we give new generalization guarantees for RERM. As future work, we are interested in understanding whether our robust loss relaxations can be used to mitigate the tradeoff between achieving adversarial robustness and maintaining high nominal performance. In addition, it is also interesting to find a combinatorial characterization of proper probabilistically robust PAC learning with respect to $\ell_{\mathcal{G},\mu}^\rho$.

## Acknowledgements

We acknowledge the support of NSF via grant IIS-2007055. We also acknowledge Idan Attias for introducing us to Tolerantly Robust PAC Learning. VR acknowledges the support of the NSF Graduate Research Fellowship.

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
