## A Equivalence between Adversarial Robustness Models

We show that the perturbation set and perturbation function models are equivalent.

**Theorem A.1** (Equivalence between $\mathcal{G}$ and $\mathcal{U}$). *Let $\mathcal{X}$ be an arbitrary domain. There exists a perturbation set $\mathcal{U} : \mathcal{X} \to 2^{\mathcal{X}}$ if and only if there exists a set of perturbation functions $\mathcal{G}$ such that $\mathcal{G}(x) = \{g(x) : g \in \mathcal{G}\} = \mathcal{U}(x)$ for all $x \in \mathcal{X}$.*

*Proof.* We first show that every set of perturbation functions $\mathcal{G}$ induces a perturbation set $\mathcal{U}$. Let $\mathcal{G}$ be an arbitrary set of perturbation functions $g : \mathcal{X} \to \mathcal{X}$. Then, for each $x \in \mathcal{X}$, define $\mathcal{U}(x) := \{g(x) : g \in \mathcal{G}\}$, which completes the proof of this direction.

Now we will show the converse - every perturbation set $\mathcal{U}$ induces a point-wise equivalent set $\mathcal{G}$ of perturbation functions. Let $\mathcal{U}$ be an arbitrary perturbation set mapping points in $\mathcal{X}$ to subsets in $\mathcal{X}$. Assume that $\mathcal{U}(x)$ is not empty for all $x \in \mathcal{X}$. Let $\tilde{z}_x$ denote an arbitrary perturbation from $\mathcal{U}(x)$. For every $x \in \mathcal{X}$, and every $z \in \mathcal{U}(x)$, define the perturbation function $g_z^x(t) = z\mathbb{1}\{t = x\} + \tilde{z}_t \mathbb{1}\{t \neq x\}$ for $t \in \mathcal{X}$. Observe that $g_z^x(x) = z \in \mathcal{U}(x)$ and $g_z^x(x') = \tilde{z}_{x'} \in \mathcal{U}(x')$. Finally, let $\mathcal{G} = \bigcup_{x \in \mathcal{X}} \bigcup_{z \in \mathcal{U}(x)} \{g_z^x\}$. To verify that $\mathcal{G} = \mathcal{U}$, consider an arbitrary point $x' \in \mathcal{X}$. Then,

$$
\begin{aligned}
\mathcal{G}(x') &= \bigcup_{x \in \mathcal{X}} \bigcup_{z \in \mathcal{U}(x)} \{g_z^x(x')\} \\
&= \left( \bigcup_{z \in \mathcal{U}(x')} \{g_z^{x'}(x')\} \right) \cup \left( \bigcup_{x \in \mathcal{X} \setminus x'} \bigcup_{z \in \mathcal{U}(x)} \{g_z^x(x')\} \right) \\
&= \left( \bigcup_{z \in \mathcal{U}(x')} \{z\} \right) \cup \left( \bigcup_{x \in \mathcal{X} \setminus x'} \bigcup_{z \in \mathcal{U}(x)} \{\tilde{z}_{x'}\} \right) \\
&= \mathcal{U}(x') \cup \tilde{z}_{x'} \\
&= \mathcal{U}(x').
\end{aligned}
$$

as needed. $\qquad\square$

## B Proofs for Section 3

### B.1 Proper $\rho$-Probabilistically Robust PAC Learning for finite $\mathcal{G}$

We show that if $\mathcal{G}$ is *finite* then VC classes are $\rho$-probabilistically robustly learnable.

**Theorem B.1** (Proper $\rho$-Probabilistically Robust PAC Learner). *For every hypothesis class $\mathcal{H}$, threshold $\rho \in [0, 1)$, perturbation set $\mathcal{G}$, and perturbation measure $\mu$ such that $|\mathcal{G}| \leq K$, there exists a proper learning rule $\mathcal{A} : (\mathcal{X} \times \mathcal{Y})^n \to \mathcal{H}$ such that for every distribution $\mathcal{D}$ over $\mathcal{X} \times \mathcal{Y}$, with probability at least $1 - \delta$ over $S \sim \mathcal{D}^n$, algorithm $\mathcal{A}$ achieves*

$$
R_{\mathcal{G},\mu}^{\rho}(\mathcal{A}(S); \mathcal{D}) \leq \inf_{h \in \mathcal{H}} R_{\mathcal{G},\mu}^{\rho}(h; \mathcal{D}) + \epsilon
$$

*with*

$$
n(\epsilon, \delta, \rho; \mathcal{H}, \mathcal{G}, \mu) = O\left( \frac{\mathrm{VC}(\mathcal{H}) \ln(K) + \ln(\frac{1}{\delta})}{\epsilon^2} \right)
$$

*samples.*

*Proof.* Fix $\rho \in (0, 1)$. Our main strategy will be to upper bound the VC dimension of the $\rho$-probabilistically robust loss class by some function of the VC dimension of $\mathcal{H}$. Then, finite VC dimension of $\mathcal{H}$ implies finite VC dimension of the loss class, which ultimately implies uniform convergence over the $\rho$-probabilistically robust loss. Finally, uniform convergence of $\ell_{\mathcal{G},\mu}^{\rho}(h, (x, y))$ implies that ERM is sufficient for $\rho$-probabilistically robust PAC learning. To that end, let

$$
\mathcal{L}_{\mathcal{G},\mu}^{\mathcal{H},\rho} = \{(x, y) \mapsto \mathbb{1}\{\mathbb{P}_{g \sim \mu}(h(g(x)) \neq y) > \rho\} : h \in \mathcal{H}\}
$$

be the $\rho$-probabilistically robust loss class of $\mathcal{H}$. Let $S = \{(x_1, y_1), ...., (x_n, y_n)\} \in (\mathcal{X} \times \mathcal{Y})^n$ be an arbitrary labeled sample of size $n$. Inflate $S$ to $S_\mathcal{G}$ by adding for each labelled example $(x, y) \in S$ all possible perturbed examples $(g(x), y)$ for $g \in \mathcal{G}$. That is, $S_\mathcal{G} = \bigcup_{(x,y) \in S} \{(g(x), y) : g \in \mathcal{G}\}$. Note that $|S_\mathcal{G}| \leq nK$. Let $\mathcal{L}^{\mathcal{H},\rho}_{\mathcal{G},\mu}(S)$ denote the set of all possible behaviors of functions in $\mathcal{L}^{\mathcal{H},\rho}_{\mathcal{G},\mu}$ on $S$. Likewise, let $\mathcal{H}(S_\mathcal{G})$ denote the set of all possible behaviors of functions in $\mathcal{H}$ on the inflated set $S_\mathcal{G}$. Note that each behavior in $\mathcal{L}^{\mathcal{H},\rho}_{\mathcal{G},\mu}(S)$ maps to at least 1 behavior in $\mathcal{H}$. Therefore $|\mathcal{L}^{\mathcal{H},\rho}_{\mathcal{G},\mu}(S)| \leq |\mathcal{H}(S_\mathcal{G})|$. By Sauer-Shelah's lemma, $|\mathcal{H}(S_\mathcal{G})| \leq (nK)^{\text{VC}(\mathcal{H})}$. Solving for $n$ such that $(nK)^{\text{VC}(\mathcal{H})} < 2^n$ gives that $n = O(\text{VC}(\mathcal{H}) \ln(K))$, ultimately implying that $\text{VC}(\mathcal{L}^{\mathcal{H},\rho}_{\mathcal{G},\mu}) \leq O(\text{VC}(\mathcal{H}) \ln(K))$ (see Lemma 1.1 in Attias et al. [2021]).

Since for VC classes, the VC dimension of $\mathcal{L}^{\mathcal{H},\rho}_{\mathcal{G},\mu}$ is bounded, by Vapnik's "General Learning", we have that for VC classes the loss function $\ell^\rho_{\mathcal{G},\mu}(h, (x, y))$ enjoys the uniform convergence property.

Namely, let $\mathcal{D}$ be a distribution over $\mathcal{X} \times \mathcal{Y}$. For a sample of size $n \geq O(\frac{\text{VC}(\mathcal{H}) \ln(K) + \ln(\frac{1}{\delta})}{\epsilon^2})$, we have that with probability at least $1 - \delta$ over $S \sim \mathcal{D}^n$, for all $h \in \mathcal{H}$

$$|\mathbb{E}_\mathcal{D}\left[\ell^\rho_{\mathcal{G},\mu}(h, (x, y))\right] - \hat{\mathbb{E}}_S\left[\ell^\rho_{\mathcal{G},\mu}(h, (x, y))\right]| \leq \epsilon.$$

Standard arguments yield that the proper learning rule $\mathcal{A}(S) = \arg\min_{h \in \mathcal{H}} \hat{\mathbb{E}}_S\left[\ell^\rho_{\mathcal{G},\mu}(h, (x, y))\right]$ is a $\rho$-probabilistically robust PAC learner with sample complexity $O(\frac{\text{VC}(\mathcal{H}) \ln(K) + \ln(\frac{1}{\delta})}{\epsilon^2})$. $\square$

### B.2 Proof of Lemma 3.2

*Proof.* Fix $\rho \in [0, 1)$ and let $m \in \mathbb{N}$. Pick $m$ center points $c_1, ..., c_m$ in $\mathcal{X}$ such that for all $i, j \in [m]$, $\mathcal{G}(c_i) \cap \mathcal{G}(c_j) = \emptyset$. For each center $c_i$, consider $2^{m-1} + 1$ disjoint subsets of its perturbation set $\mathcal{G}(c_i)$ which do not contain $c_i$. Label $2^{m-1}$ of these subsets with a unique bitstring $b \in \{0, 1\}^m$ fixing $b_i = 1$. Let $\mathcal{B}^b_i$ denote the subset labeled by bitstring $b$ and let $\mathcal{B}_i$ denote the single remaining subset that was not labeled. Furthermore, for each $i \in [m]$ and $b \in \{\{0, 1\}^m | b_i = 1\}$, pick $\mathcal{B}_i$ and $\mathcal{B}^b_i$'s such that $\mu_{c_i}(\mathcal{B}_i) = \rho$ and $0 < \mu_{c_i}(\mathcal{B}^b_i) \leq \frac{1-\rho}{2^m}$. If $b_i = 0$, let $\mathcal{B}^b_i = \emptyset$. If $\rho = 0$, let $\mathcal{B}_i = \emptyset$ for all $i \in [m]$. Finally, define $\mathcal{B} = \bigcup_{i=1}^m \bigcup_{b \in \{0,1\}^m} \mathcal{B}^b_i \cup \mathcal{B}_i$ as the union of all the subsets. Crucially, observe that for all $i \in [m]$, $\mu_{c_i}\left(\mathcal{B}_i \cup \left(\bigcup_b \mathcal{B}^b_i\right)\right) \leq \frac{1+\rho}{2} < 1$.

For bitstring $b \in \{0, 1\}^m$, define the hypothesis $h_b$ as

$$h_b(z) = \begin{cases} -1 & \text{if } z \in \bigcup_{i=1}^m \mathcal{B}^b_i \cup \mathcal{B}_i \\ 1 & \text{otherwise} \end{cases}$$

and consider the hypothesis class $\mathcal{H} = \{h_b | b \in \{0, 1\}^m\}$ which consists of all $2^m$ hypothesis, one for each bitstring. We first show that $\mathcal{H}$ has VC dimension at most 1. Consider two points $x_1, x_2 \in \mathcal{X}$. We will show case by case that every possible pair of points cannot be shattered by $\mathcal{H}$. First, consider the case where, wlog, $x_1 \notin \mathcal{B}$. Then, $\forall h \in \mathcal{H}$, $h(x_1) = 1$, and thus shattering is not possible. Now, consider the case where both $x_1 \in \mathcal{B}$ and $x_2 \in \mathcal{B}$. If either $x_1$ or $x_2$ is in $\bigcup_{i=1}^m \mathcal{B}_i$, then every hypothesis $h \in \mathcal{H}$ will label it as $-1$, and thus these two points cannot be shattered. If $x_1 \in \mathcal{B}^b_i$ and $x_2 \in \mathcal{B}^b_j$ for $i \neq j$, then $h_b(x_1) = h_b(x_2) = -1$, but $\forall h \in \mathcal{H}$ such that $h \neq h_b$, $h(x_1) = h(x_2) = 1$. If $x_1 \in \mathcal{B}^{b_1}_i$ and $x_2 \in \mathcal{B}^{b_2}_j$ for $b_1 \neq b_2$, then there exists no hypothesis in $\mathcal{H}$ that can label $(x_1, x_2)$ as $(-1, -1)$. Thus, overall, no two points $x_1, x_2 \in \mathcal{X}$ can be shattered by $\mathcal{H}$.

Now we are ready to show that the VC dimension of the loss class is at least $m$. Specifically, given the sample of labelled points $S = \{(c_1, 1), ..., (c_m, 1)\}$, we will show that the loss behavior corresponding to hypothesis $h_b$ on the sample $S$ is exactly $b$. Since $\mathcal{H}$ contains all the hypothesis corresponding to every single bitstring $b \in \{0, 1\}^m$, the loss class of $\mathcal{H}$ will shatter $S$. In order to prove that the loss behavior of $h_b$ on the sample $S$ is exactly $b$, it suffices to show that the probabilistic

loss of $h_b$ on example $(c_i, 1)$ is $b_i$, where $b_i$ denotes the $i$th bit of $b$. By definition,

$$
\begin{aligned}
\ell_{\mathcal{G},\mu}^{\rho}(h_b, (c_i, 1)) &= \mathbb{1}\{\mathbb{P}_{g\sim\mu}\left(h_b(g(c_i)) \neq 1\right) > \rho\} \\
&= \mathbb{1}\{\mathbb{P}_{z\sim\mu_{c_i}}\left(h_b(z) = 0\right) > \rho\} \\
&= \mathbb{1}\{\mathbb{P}_{z\sim\mu_{c_i}}\left(z \in \mathcal{B}_i^b \cup \mathcal{B}_i\right) > \rho\} \\
&= \mathbb{1}\{\mu_{c_i}(\mathcal{B}_i^b \cup \mathcal{B}_i) > \rho\} \\
&= b_i.
\end{aligned}
$$

Thus, the loss behavior of $h_b$ on $S$ is $b$, and the total number of distinct loss behaviors over each hypothesis in $\mathcal{H}$ on $S$ is $2^m$, implying that the VC dimension of the loss class is at least $m$. This completes the construction and proof of the claim. $\qquad\square$

### B.3  Proof of Lemma 3.3

*Proof.* (of Lemma 3.3) This proof closely follows Lemma 3 from Montasser et al. [2019]. In fact, the only difference is in the construction of the hypothesis class, which we will describe below.

Fix $\rho \in [0, 1)$. Let $m \in \mathbb{N}$. Construct a hypothesis class $\mathcal{H}_0$ as in Lemma 3.2 on $3m$ centers $c_1, ..., c_{3m}$ based on $\rho$. By the construction in Lemma 3.2, we know that $\mathcal{L}_{\mathcal{G},\mu}^{\mathcal{H},\rho}$ shatters the sample $C = \{(c_1, 1), ..., (c_{3m}, 1)\}$. Instead of keeping all of $\mathcal{H}_0$, we will only keep a subset $\mathcal{H}$ of $\mathcal{H}_0$, namely those classifiers that are probabilistically robustly correct on subsets of size $2m$ of $C$. More specifically, recall from the construction in Lemma 3.2, that each hypothesis $h_b \in \mathcal{H}_0$ is parameterized by a bitstring $b \in \{0, 1\}^{3m}$ where if $b_i = 1$, then $h_b$ is not robust to example $(c_i, 1)$. Therefore, $\mathcal{H} = \{h_b \in \mathcal{H}_0 : \sum_{i=1}^{3m} b_i = m\}$. Now, let $\mathcal{A} : (\mathcal{X} \times \mathcal{Y})^* \to \mathcal{H}$ be an arbitrary proper learning rule. Consider a set of distributions $\mathcal{D}_1, ..., \mathcal{D}_L$ where $L = \binom{3m}{2m}$. Each distribution $\mathcal{D}_i$ is uniform over exactly $2m$ centers in $C$. Critically, note that by our construction of $\mathcal{H}$, every distribution $\mathcal{D}_i$ is probabilistically robustly realizable by a hypothesis in $\mathcal{H}$. That is, for all $\mathcal{D}_i$, there exists a hypothesis $h^* \in \mathcal{H}$ such that $R_{\mathcal{G},\mu}^{\rho}(h^*; \mathcal{D}_i) = 0$. Observe that this satisfies the first condition in Lemma 3.3. For the second condition, at a high-level, the idea is to use the probabilistic method to show that there exists a distribution $\mathcal{D}_i$ where $\mathbb{E}_{S\sim\mathcal{D}_i^m}\left[R_{\mathcal{G},\mu}^{\rho}(\mathcal{A}(S); \mathcal{D})\right] \geq \frac{1}{4}$ and then use a variant of Markov's inequality to show that with probability at least $1/7$ over $S \sim \mathcal{D}^m$, $R_{\mathcal{G},\mu}^{\rho}(\mathcal{A}(S); \mathcal{D}) > 1/8$.

Let $S \in C^m$ be an arbitrary set of $m$ points. Let $\mathcal{C}$ be a uniform distribution over $C$. Let $\mathcal{P}$ be a uniform distribution over $\mathcal{D}_1, ..., \mathcal{D}_T$. Let $E_S$ denote the event that $S \subset \text{supp}(\mathcal{D}_i)$ for $\mathcal{D}_i \sim \mathcal{P}$. Given the event $E_S$, we will lower bound the expected probabilistic robust loss of the hypothesis the proper learning rule $\mathcal{A}$ outputs,

$$
\mathbb{E}_{\mathcal{D}_i\sim\mathcal{P}}\left[R_{\mathcal{G},\mu}^{\rho}(\mathcal{A}(S); \mathcal{D}_i)|E_S\right] = \mathbb{E}_{\mathcal{D}_i\sim\mathcal{P}}\left[\mathbb{E}_{(x,y)\sim\mathcal{D}_i}\left[\mathbb{1}\{\mathbb{P}_{g\sim\mu}\left(\mathcal{A}(S)(g(x)) \neq y\right) > \rho\}\right]|E_S\right].
$$

Conditioning on the event that $(x, y) \notin S$, denoted, $E_{(x,y)\notin S}$,

$$
\begin{aligned}
\mathbb{E}_{(x,y)\sim\mathcal{D}_i}\left[\mathbb{1}\{\mathbb{P}_{g\sim\mu}\left(\mathcal{A}(S)(g(x)) \neq y\right) > \rho\}\right] \geq \; &\mathbb{P}_{(x,y)\sim\mathcal{D}_i}\left[E_{(x,y)\notin S}\right] \\
&\times \mathbb{E}_{(x,y)\sim\mathcal{D}_i}\left[\mathbb{1}\{\mathbb{P}_{g\sim\mu}\left(\mathcal{A}(S)(g(x)) \neq y\right) > \rho\}|E_{(x,y)\notin S}\right]
\end{aligned}
$$

Since $\mathcal{D}_i$ is supported over $2m$ points and $|S| = m$, $\mathbb{P}_{(x,y)\sim\mathcal{D}_i}\left[E_{(x,y)\notin S}\right] \geq \frac{1}{2}$ since in the worst-case $S \subset \text{supp}(\mathcal{D}_i)$. Thus, we obtain the lower bound,

$$
\mathbb{E}_{\mathcal{D}_i\sim\mathcal{P}}\left[R_{\mathcal{G},\mu}^{\rho}(\mathcal{A}(S); \mathcal{D}_i)|E_S\right] \geq \frac{1}{2}\mathbb{E}_{\mathcal{D}_i\sim\mathcal{P}}\left[\mathbb{E}_{(x,y)\sim\mathcal{D}_i}\left[\mathbb{1}\{\mathbb{P}_{g\sim\mu}\left(\mathcal{A}(S)(g(x)) \neq y\right) > \rho\}|E_{(x,y)\notin S}\right]|E_S\right].
$$

Unravelling the expectation over the draw from $\mathcal{D}_i$ given the event $E_S$, we have,

$$\mathbb{E}_{(x,y)\sim\mathcal{D}_i}\left[\mathbb{1}\{\mathbb{P}_{g\sim\mu}\left(\mathcal{A}(S)(g(x))\neq y\right)>\rho\}|E_{(x,y)\notin S}\right]\geq\frac{1}{m}\sum_{(x,y)\in\mathrm{supp}(\mathcal{D}_i)\setminus S}\mathbb{1}\{\mathbb{P}_{g\sim\mu}\left(\mathcal{A}(S)(g(x))\neq y\right)>\rho\}$$

Observing that $\mathbb{E}_{\mathcal{D}_i\sim\mathcal{P}}\left[\mathbb{1}\{(x,y)\in\mathrm{supp}(\mathcal{D}_i)\}|E_S\right]\geq\frac{1}{2}$ yields,

$$\mathbb{E}_{\mathcal{D}_i\sim\mathcal{P}}\left[\mathbb{E}_{(x,y)\sim\mathcal{D}_i}\left[\mathbb{1}\{\mathbb{P}_{g\sim\mu}\left(\mathcal{A}(S)(g(x))\neq y\right)>\rho\}|E_{(x,y)\notin S}\right]|E_S\right]\geq\frac{1}{2m}\sum_{(x,y)\notin S}\mathbb{1}\{\mathbb{P}_{g\sim\mu}\left(\mathcal{A}(S)(g(x))\neq y\right)>\rho\}.$$

Since $\mathcal{A}(S)\in\mathcal{H}$, by construction of $\mathcal{H}$, there are at least $m$ points in $C$ where $\mathcal{A}$ is not probabilistically robustly correct. Therefore,

$$\frac{1}{2m}\sum_{(x,y)\notin S}\mathbb{1}\{\mathbb{P}_{g\sim\mu}\left(\mathcal{A}(S)(g(x))\neq y\right)>\rho\}\geq\frac{1}{2},$$

from which we have that, $\mathbb{E}_{\mathcal{D}_i\sim\mathcal{P}}\left[R_{\mathcal{G},\mu}^\rho(\mathcal{A}(S);\mathcal{D}_i)|E_S\right]\geq\frac{1}{4}$. By the law of total expectation, we have that

$$\mathbb{E}_{\mathcal{D}_i\sim\mathcal{P}}\left[\mathbb{E}_{S\sim\mathcal{D}_i^m}\left[R_{\mathcal{G},\mu}^\rho(\mathcal{A}(S);\mathcal{D}_i)\right]\right]=\mathbb{E}_{S\sim C}\left[\mathbb{E}_{\mathcal{D}_i\sim\mathcal{P}|E_S}\left[R_{\mathcal{G},\mu}^\rho(\mathcal{A}(S);\mathcal{D}_i)\right]\right]$$
$$=\mathbb{E}_{S\sim C}\left[\mathbb{E}_{\mathcal{D}_i\sim\mathcal{P}}\left[R_{\mathcal{G},\mu}^\rho(\mathcal{A}(S);\mathcal{D}_i)|E_S\right]\right]$$
$$\geq 1/4$$

Since the expectation over $\mathcal{D}_1,...,\mathcal{D}_T$ is at least $1/4$, there must exist a distribution $\mathcal{D}_i$ where $\mathbb{E}_{S\sim\mathcal{D}_i^m}\left[R_{\mathcal{G},\mu}^\rho(\mathcal{A}(S);\mathcal{D}_i)\right]\geq 1/4$. Using a variant of Markov's inequality, gives

$$\mathbb{P}_{S\sim\mathcal{D}_i^m}\left[R_{\mathcal{G},\mu}^\rho(\mathcal{A}(S);\mathcal{D}_i)>1/8\right]\geq 1/7$$

which completes the proof. $\qquad\square$

## B.4 Proof of Theorem 3.1

*Proof.* (of Theorem 3.1) Fix $\rho\in[0,1)$. Let $(C_m)_{m\in\mathbb{N}}$ be an infinite sequence of disjoint sets such that each set $C_m$ contains $3m$ distinct center points from $\mathcal{X}$, where for any $c_i,c_j\in\bigcup_{m=1}^\infty C_m$ such that $c_i\neq c_j$, we have $\mathcal{G}(c_i)\cap\mathcal{G}(c_j)=\emptyset$. For every $m\in\mathbb{N}$, construct $\mathcal{H}_m$ on $C_m$ as in Lemma 3.2. In addition, a key part of this proof is to ensure that the hypothesis in $\mathcal{H}_m$ are non-robust to points in $C_{m'}$ for all $m'\neq m$. To do so, we will need to adjust each hypothesis $h_b\in\mathcal{H}_m$ carefully. By definition, for every $m\in\mathbb{N}$, $\mathcal{H}_m$ consists of $2^{3m}$ hypothesis of the form

$$h_b(z)=\begin{cases}-1 & \text{if }z\in\bigcup_{i=1}^{3m}\mathcal{B}_i^b\cup\mathcal{B}_i\\1 & \text{otherwise}\end{cases}$$

for each bitstring $b\in\{0,1\}^{3m}$. Note that the same set $\bigcup_{i=1}^{3m}\mathcal{B}_i$ is shared across every hypothesis $h_b\in\mathcal{H}_m$. For each $m\in\mathbb{N}$, let $\mathcal{B}^m=\bigcup_{i=1}^{3m}\mathcal{B}_i$ be exactly the union of these $3m$ sets. Next, from the construction in Lemma 3.2, for every center $c_i\in C_m$, $\mu_{c_i}\left(\mathcal{B}_i\cup\left(\bigcup_b\mathcal{B}_i^b\right)\right)\leq\frac{1+\rho}{2}<1$. Thus, there exists a set $\tilde{\mathcal{B}}_i\subset\mathcal{G}(c_i)$ such that $\mu_{c_i}(\tilde{\mathcal{B}}_i)>0$ and $\tilde{\mathcal{B}}_i\cap\left(\mathcal{B}_i\cup\left(\bigcup_b\mathcal{B}_i^b\right)\right)=\emptyset$. Consider one such subset $\tilde{\mathcal{B}}_i$ from each of the $3m$ centers in $C_m$ and let $\tilde{\mathcal{B}}^m=\bigcup_{i=1}^{3m}\tilde{\mathcal{B}}_i$. Finally, make the following adjustment to each $h_b\in\mathcal{H}_m$,

$$h_b(z)=\begin{cases}-1 & \text{if }z\in\bigcup_{i=1}^{3m}\mathcal{B}_i^b\cup\mathcal{B}_i\text{ or }z\in\mathcal{B}^{m'}\cup\tilde{\mathcal{B}}^{m'}\text{ for }m'\neq m\\1 & \text{otherwise}\end{cases}$$

One can verify that every hypothesis in $\mathcal{H}_m$ has a non-robust region (i.e. $\mathcal{B}^{m'} \cup \tilde{\mathcal{B}}^{m'}$ for $m' \neq m$) with mass strictly bigger than $\rho$ in every center in $C_{m'}$ for every $m' \neq m$. Thus, the hypotheses in $\mathcal{H}_m$ are non-robust to points in $C_{m'}$ for all $m' \neq m$. Finally, as we did in Lemma 3.3, for each $m$, we only keep the subset of hypothesis $\mathcal{H}'_m = \{h_b \in \mathcal{H}_m : \sum_{i=1}^{3m} b_i = m\}$. Note that for each $m \in \mathbb{N}$, the hypothesis class $\mathcal{H}'_m$ behaves exactly like the hypothesis class from Lemma 3.3 on $C_m$.

Let $\mathcal{H} := \bigcup_{m=1}^{\infty} \mathcal{H}'_m$ and $\mathcal{G}(C_m) := \bigcup_{i=1}^{3m} \mathcal{G}(c_i)$. Since we have modified the hypothesis class, we need to reprove that its VC dimension is still at most 1. Consider two points $x_1, x_2 \in \mathcal{X}$. If either $x_1$ or $x_2$ is not in $\bigcup_{m=1}^{\infty} \mathcal{G}(C_m)$ and not in $\bigcup_{m=1}^{\infty} \mathcal{B}^m \cup \tilde{\mathcal{B}}^m$, then all hypothesis predict $x_1$ or $x_2$ as 1. If both $x_1$ and $x_2$ are in $\mathcal{B}^m \cup \tilde{\mathcal{B}}^m$ for some $m \in \mathbb{N}$, then:

- if either $x_1$ or $x_2$ are in $\mathcal{B}^m$, every hypothesis in $\mathcal{H}$ labels either $x_1$ or $x_2$ as $-1$.
- if both $x_1$ and $x_2$ are in $\tilde{\mathcal{B}}^m$, we can only get the labeling $(1,1)$ from hypotheses in $\mathcal{H}_m$ and the labeling $(-1,-1)$ from the hypotheses in $\mathcal{H}_{m'}$ for $m' \neq m$.

In the case both $x_1$ and $x_2$ are in $\mathcal{G}(C_m) \setminus (\mathcal{B}^m \cup \tilde{\mathcal{B}}^m)$, then, they cannot be shattered by Lemma 3.2. In the case $x_1 \in \mathcal{B}^m \cup \tilde{\mathcal{B}}^m$ and $x_2 \in \mathcal{G}(C_m) \setminus (\mathcal{B}^m \cup \tilde{\mathcal{B}}^m)$:

- if $x_1$ is in $\mathcal{B}^m$, every hypothesis in $\mathcal{H}$ labels $x_1$ as $-1$.
- if $x_1$ is in $\tilde{\mathcal{B}}^m$ then, we can never get the labeling $(-1, -1)$.

If $x_1 \in \mathcal{B}^i \cup \tilde{\mathcal{B}}^i$ and $x_2 \in \mathcal{B}^j \cup \tilde{\mathcal{B}}^j$ for $i \neq j$, then:

- if either $x_1$ or $x_2$ are in $\mathcal{B}^i$ or $\mathcal{B}^j$ respectively, every hypothesis in $\mathcal{H}$ labels either $x_1$ or $x_2$ as $-1$.
- if both $x_1$ and $x_2$ are in $\tilde{\mathcal{B}}^i$ and $\tilde{\mathcal{B}}^j$ respectively, we can never get the labeling $(1,1)$.

In the case $x_1 \in \mathcal{B}^i \cup \tilde{\mathcal{B}}^i$ and $x_2 \in \mathcal{G}(C_j) \setminus (\mathcal{B}^j \cup \tilde{\mathcal{B}}^j)$ for $j \neq i$, then we cannot obtain the labeling $(1, -1)$. If $x_1 \in \mathcal{G}(C_i) \setminus (\mathcal{B}^i \cup \tilde{\mathcal{B}}^i)$ and $x_2 \in \mathcal{G}(C_j) \setminus (\mathcal{B}^j \cup \tilde{\mathcal{B}}^j)$ for $i \neq j$, then we cannot obtain the labeling $(-1, -1)$. Since we shown that for all possible $x_1$ and $x_2$, $\mathcal{H}$ cannot shatter them, $\text{VC}(\mathcal{H}) \leq 1$.

We now use the same reasoning in Montasser et al. [2019], to show that no proper learning rule works. By Lemma 3.3, for any proper learning rule $\mathcal{A} : (\mathcal{X} \times \mathcal{Y})^* \to \mathcal{H}$ and for any $m \in \mathbb{N}$, we can construct a distribution $\mathcal{D}$ over $C_m$ (which has $3m$ points from $\mathcal{X}$) where there exists a hypothesis $h^* \in \mathcal{H}'_m$ that achieves $R^\rho_{\mathcal{G},\mu}(h^*; \mathcal{D}) = 0$, but with probability at least $1/7$ over $S \sim \mathcal{D}^m$, $R^\rho_{\mathcal{G},\mu}(\mathcal{A}(S); \mathcal{D}) > 1/8$. Note that it suffices to only consider hypothesis in $\mathcal{H}'_m$ because, by construction, all hypothesis in $\mathcal{H}'_{m'}$ for $m' \neq m$ are not probabilistically robust on $C_m$, and thus always achieve loss 1 on all points in $C_m$. Thus, rule $\mathcal{A}$ will do worse if it picks hypotheses from these classes. This shows that the sample complexity of properly probabilistically robustly PAC learning $\mathcal{H}$ is arbitrarily large, allowing us to conclude that $\mathcal{H}$ is not properly learnable. $\qquad \square$

## C Proofs for Section 4

### C.1 Proof of Theorem 4.2

*Proof.* (of Theorem 4.2) Let $\text{VC}(\mathcal{H}) = d$ and $S = \{(x_1, y_1), ..., (x_m, y_m)\}$ an i.i.d. sample of size $m$ from $\mathcal{D}$. Consider the learning algorithm $\mathcal{A}(S) = \arg\min_{h \in \mathcal{H}} \hat{\mathbb{E}}_S [\ell_{\mathcal{G},\mu}(h, (x, y))]$. Note that $\mathcal{A}$ is a proper learning algorithm. Let $\hat{h} = \mathcal{A}(S)$ denote hypothesis output by $\mathcal{A}$ and $h^* = \inf_{h \in \mathcal{H}} \mathbb{E}_\mathcal{D} [\ell_{\mathcal{G},\mu}(h, (x, y))]$.

We now show that if the sample size $m = O\left(\frac{dL^2 \ln(\frac{L}{\epsilon}) + \ln(\frac{1}{\delta})}{\epsilon^2}\right)$, then $\hat{h}$ achieves the stated generalization bound with probability $1 - \delta$. By Lemma 4.1, if $m = O\left(\frac{dL^2 \ln(\frac{L}{\epsilon}) + \ln(\frac{1}{\delta})}{\epsilon^2}\right)$, we have that

with probability $1 - \delta$, for all $h \in \mathcal{H}$ simultaneously,

$$\left| \mathbb{E}_{\mathcal{D}} \left[ \ell_{\mathcal{G},\mu}(h, (x,y)) \right] - \hat{\mathbb{E}}_S \left[ \ell_{\mathcal{G},\mu}(h, (x,y)) \right] \right| \leq \frac{\epsilon}{2}.$$

This means that both $\mathbb{E}_{\mathcal{D}} \left[ \ell_{\mathcal{G},\mu}(\hat{h}, (x,y)) \right] - \hat{\mathbb{E}}_S \left[ \ell_{\mathcal{G},\mu}(\hat{h}, (x,y)) \right] \leq \frac{\epsilon}{2}$ and $\hat{\mathbb{E}}_S \left[ \ell_{\mathcal{G},\mu}(h^*, (x,y)) \right] - \mathbb{E}_{\mathcal{D}} \left[ \ell_{\mathcal{G},\mu}(h^*, (x,y)) \right] \leq \frac{\epsilon}{2}$. By definition of $\hat{h}$, note that $\hat{\mathbb{E}}_S \left[ \ell_{\mathcal{G},\mu}(\hat{h}, (x,y)) \right] \leq \hat{\mathbb{E}}_S \left[ \ell_{\mathcal{G},\mu}(h^*, (x,y)) \right]$. Putting these observations together, we have that

$$\mathbb{E}_{\mathcal{D}} \left[ \ell_{\mathcal{G},\mu}(\hat{h}, (x,y)) \right] - \left( \mathbb{E}_{\mathcal{D}} \left[ \ell_{\mathcal{G},\mu}(h^*, (x,y)) \right] + \frac{\epsilon}{2} \right) \leq \mathbb{E}_{\mathcal{D}} \left[ \ell_{\mathcal{G},\mu}(\hat{h}, (x,y)) \right] - \hat{\mathbb{E}}_S \left[ \ell_{\mathcal{G},\mu}(h^*, (x,y)) \right]$$
$$\leq \mathbb{E}_{\mathcal{D}} \left[ \ell_{\mathcal{G},\mu}(\hat{h}, (x,y)) \right] - \hat{\mathbb{E}}_S \left[ \ell_{\mathcal{G},\mu}(\hat{h}, (x,y)) \right]$$
$$\leq \frac{\epsilon}{2},$$

from which we can deduce that

$$\mathbb{E}_{\mathcal{D}} \left[ \ell_{\mathcal{G},\mu}(\hat{h}, (x,y)) \right] - \inf_{h \in \mathcal{H}} \mathbb{E}_{\mathcal{D}} \left[ \ell_{\mathcal{G},\mu}(h, (x,y)) \right] \leq \epsilon.$$

Thus, $\mathcal{A}$ achieves the stated generalization bound with sample complexity $m = O\left( \frac{dL^2 \ln(\frac{L}{\epsilon}) + \ln(\frac{1}{\delta})}{\epsilon^2} \right)$, completing the proof. $\qquad \square$

## C.2 Proof of Theorem 4.3

For the proof in this section, it will be useful to define the $(\mathcal{G}, \mu)$-smoothed hypothesis class $\mathcal{H}$:

$$\mathcal{F}_{\mathcal{G},\mu}^{\mathcal{H}} := \{ \mathbb{E}_{g \sim \mu} [h(g(x))] : h \in \mathcal{H} \}.$$

*Proof.* (of Theorem 4.3) Let $\mathcal{X} = \mathbb{R}$ and $\mathcal{H} = \{\text{sign}(\sin(\omega x)) : \omega \in \mathbb{R}\}$. Without loss of generality, assume $\text{sign}(\sin(0)) = 1$. For every $x \in \mathcal{X}$ and $c \in [-1, 1]$, define $g_c(x) = cx$. Then, let $\mathcal{G} = \{g_c : c \in [-1, 1]\}$ and $\mu$ be uniform over $\mathcal{G}$. First, $VC(\mathcal{H}) = \infty$ as desired. Next, to show learnability, it suffices to show that the loss

$$\ell_{\mathcal{G},\mu}(h, (x,y)) = \ell(y \mathbb{E}_{g \sim \mu} [h(g(x))]).$$

enjoys the uniform convergence property despite $VC(\mathcal{H}) = \infty$. By Theorem 2.1 and similar to the proof of Lemma 4.1, it suffices upperbound the Rademacher complexity of the loss class $\mathcal{L}_{\mathcal{G},\mu}^{\mathcal{H}} = \{(x,y) \mapsto \ell_{\mathcal{G},\mu}(h, (x,y)) : h \in \mathcal{H}\}$. Since for every fixed $y$, $\ell_{\mathcal{G},\mu}(h, (x,y))$ is $L$-Lipschitz with respect to the real-valued function $\mathbb{E}_{g \sim \mu} [h(g(x))]$, by Ledoux-Talagrand's contraction principle $\hat{\mathfrak{R}}_m(\mathcal{L}_{\mathcal{G},\mu}^{\mathcal{H}}) \leq L \cdot \hat{\mathfrak{R}}_m(\mathcal{F}_{\mathcal{G},\mu}^{\mathcal{H}})$ where $\mathcal{F}_{\mathcal{G},\mu}^{\mathcal{H}}$ is the $(\mathcal{G}, \mu)$-smoothed hypothesis classed defined previously. Thus, it suffices to upper-bound $\hat{\mathfrak{R}}_m(\mathcal{F}_{\mathcal{G},\mu}^{\mathcal{H}})$ by a sublinear function of $m$ to show that $\ell_{\mathcal{G},\mu}(h, (x,y))$ enjoys the uniform convergence property. But for every $h_\omega \in \mathcal{H}$,

$$\mathbb{E}_{g \sim \mu} [h_\omega(g(x))] = \mathbb{E}_{c \sim \text{Unif}(-1,1)} [\text{sign}(\sin(\omega(cx)))] = \frac{1}{2} \int_{-1}^{1} \text{sign}(\sin(c(\omega x))) dc.$$

Since $\sin(ax)$ is an odd function, $\text{sign}(\sin(ax))$ is also odd, from which it follows that for all $h_\omega \in \mathcal{H}$:

$$\mathbb{E}_{g \sim \mu} [h_\omega(g(x))] = \begin{cases} 0 & \text{if } x \neq 0 \text{ and } \omega \neq 0 \\ 1 & \text{otherwise} \end{cases}.$$

Therefore, $\mathcal{F}_{\mathcal{G},\mu}^{\mathcal{H}} = \{f_1, f_2\}$ where $f_1(x) = 1$ for all $x \in \mathbb{R}$ and $f_2(x) = 1$ if $x = 0$ and $f_2(x) = 0$ if $x \neq 0$. Since $\mathcal{F}_{\mathcal{G},\mu}^{\mathcal{H}}$ is finite, by Massart's Lemma [Mohri et al., 2018], $\hat{\mathfrak{R}}_m(\mathcal{F}_{\mathcal{G},\mu}^{\mathcal{H}})$ is upper-bounded by a sublinear function of $m$ such that $\ell_{\mathcal{G},\mu}(h, (x,y))$ enjoys the uniform convergence property with sample complexity $O(\frac{L^2 + \ln(\frac{1}{\delta})}{\epsilon^2})$. Therefore, $(\mathcal{H}, \mathcal{G}, \mu)$ is PAC learnable with respect to $\ell_{\mathcal{G},\mu}(h, (x,y))$ by the learning rule $\mathcal{A}(S) = \arg\min_{h \in \mathcal{H}} \hat{\mathbb{E}}_S [\ell_{\mathcal{G},\mu}(h, (x,y))]$ with sample complexity that scales according to $O(\frac{L^2 + \ln(\frac{1}{\delta})}{\epsilon^2})$. $\qquad \square$

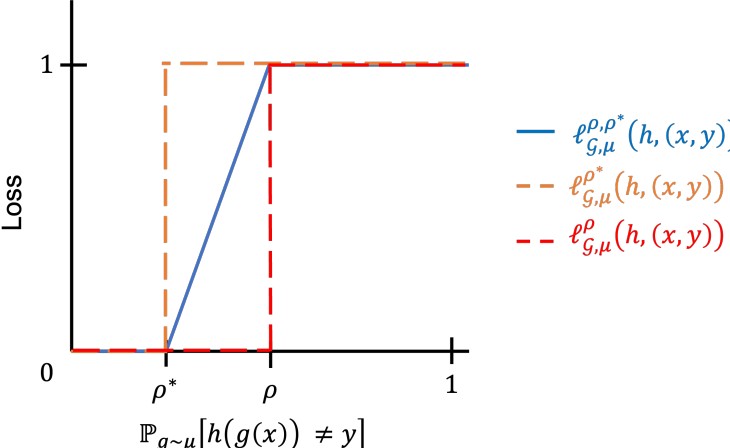

Figure 1: Comparison of probabilistic robust *ramp* loss to probabilistic robust losses of hypothesis $h$ on example $(x, y)$. The probabilistic robust losses at $\rho$ and $\rho^*$ sandwich the probabilistic robust ramp loss at $\rho, \rho^*$.

## D  Proofs for Section 5

### D.1  Proof of Theorem 5.2

*Proof.* (of Theorem 5.2) Fix $0 \leq \rho^* < \rho < 1$ and let $\mathcal{H}$ be a hypothesis class with $\text{VC}(\mathcal{H}) = d$. Let $(\mathcal{G}, \mu)$ be an arbitrary perturbation set and measure, $\mathcal{D}$ be an arbitrary distribution over $\mathcal{X} \times \mathcal{Y}$, and $S = \{(x_1, y_1), ..., (x_m, y_m)\}$ an i.i.d. sample of size $m$. Let $\mathcal{A}(S) = \text{PRERM}(S; (\mathcal{G}, \mu), \rho^*)$.

By Lemma 5.1, it suffices to show that there exists a loss function $\ell(h, (x, y))$ such that $\ell_{\mathcal{G},\mu}^{\rho}(h, (x, y)) \leq \ell(h, (x, y)) \leq \ell_{\mathcal{G},\mu}^{\rho^*}(h, (x, y)))$ and $\ell(h, (x, y))$ enjoys the uniform convergence property with sample complexity $n = O\left( \frac{\frac{d}{(\rho-\rho^*)^2} \ln(\frac{1}{(\rho-\rho^*)\epsilon}) + \ln(\frac{1}{\delta})}{\epsilon^2} \right)$. Consider the probabilistically robust ramp loss:

$$\ell_{\mathcal{G},\mu}^{\rho,\rho^*}(h, (x, y)) = \min(1, \max(0, \frac{\mathbb{P}_{g\sim\mu}\left[h(g(x)) \neq y\right] - \rho^*}{\rho - \rho^*})).$$

Figure 1 visually showcases how the probabilistic robust losses at $\rho$ and $\rho^*$ sandwich the probabilistic ramp loss at $\rho, \rho^*$.

Its not too hard to see that $\ell_{\mathcal{G},\mu}^{\rho}(h, (x, y)) \leq \ell_{\mathcal{G},\mu}^{\rho,\rho^*}(h, (x, y)) \leq \ell_{\mathcal{G},\mu}^{\rho^*}(h, (x, y)))$. Furthermore, since $\ell_{\mathcal{G},\mu}^{\rho,\rho^*}(h, (x, y))$ is $O(\frac{1}{\rho-\rho^*})$-Lipschitz in $y\mathbb{E}_{g\sim\mu}\left[h(g(x)) \neq y\right]$, by Lemma 4.1, we have that $\ell_{\mathcal{G},\mu}^{\rho,\rho^*}(h, (x, y))$ enjoys the uniform convergence property with sample complexity $O\left( \frac{\frac{d}{(\rho-\rho^*)^2} \ln(\frac{1}{(\rho-\rho^*)\epsilon}) + \ln(\frac{1}{\delta})}{\epsilon^2} \right)$. This completes the proof, as the conditions for Lemma 5.1 have been met, and therefore the learning rule $\mathcal{A}(S) = \text{PRERM}(S; \mathcal{G}, \rho^*)$ enjoys the stated generalization guarantee with the specified sample complexity. $\qquad \square$

### D.2  Proof of Theorem 5.3

*Proof.* (of Theorem 5.3) Fix $0 < \rho$ and let $\mathcal{H}$ be a hypothesis class with $\text{VC}(\mathcal{H}) = d$. Let $\mathcal{G}$ be an arbitrary perturbation set, $\mathcal{D}$ be an arbitrary distribution over $\mathcal{X} \times \mathcal{Y}$, and $S = \{(x_1, y_1), ..., (x_m, y_m)\}$ an i.i.d. sample of size $m$. Let $\mathcal{A}(S) = \text{RERM}(S; \mathcal{G})$.

Fix a measure $\mu$ over $\mathcal{G}$. By Lemma 5.1, it suffices to show that there exists a loss function $\ell(h, (x, y))$ such that $\ell_{\mathcal{G},\mu}^{\rho}(h, (x, y)) \leq \ell(h, (x, y)) \leq \ell_{\mathcal{G}}(h, (x, y)))$ and $\ell(h, (x, y))$ enjoys the

uniform convergence property with sample complexity $n = O\left(\frac{\frac{d}{\rho^2}\ln(\frac{1}{\rho\epsilon})+\ln(\frac{1}{\delta})}{\epsilon^2}\right)$. Recall the probabilistically robust ramp loss:

$$\ell_{\mathcal{G},\mu}^{\rho,\rho^*}(h,(x,y)) = \min(1, \max(0, \frac{\mathbb{P}_{g\sim\mu}\left[h(g(x))\neq y\right]-\rho^*}{\rho-\rho^*})).$$

Letting $\rho^* = 0$, its not too hard to see that $\ell_{\mathcal{G},\mu}^{\rho}(h,(x,y)) \leq \ell_{\mathcal{G},\mu}^{\rho,0}(h,(x,y)) \leq \ell_{\mathcal{G}}(h,(x,y)))$. Furthermore, since $\ell_{\mathcal{G},\mu}^{\rho,0}(h,(x,y))$ is $O(\frac{1}{\rho})$-Lipschitz in $y\mathbb{E}_{g\sim\mu}\left[h(g(x))\neq y\right]$, by Lemma 4.1, we have that $\ell_{\mathcal{G},\mu}^{\rho,0}(h,(x,y))$ enjoys the uniform convergence property with sample complexity $O\left(\frac{\frac{d}{\rho^2}\ln(\frac{1}{\rho\epsilon})+\ln(\frac{1}{\delta})}{\epsilon^2}\right)$. This completes the proof, as the conditions for Lemma 5.1 have been met, and therefore the learning rule $\mathcal{A}(S)$ enjoys the stated generalization guarantee with the specified sample complexity. $\qquad\square$

### D.3 Proof of Theorem 5.4

*Proof.* (of Theorem 5.4) Assume that there exists a subset $\mathcal{G}' \subset \mathcal{G}$, that is $r$-Nice with respect to $\mathcal{H}$. By Lemma 5.1, it is sufficient to find a perturbation set $\tilde{\mathcal{G}}$ such that (1) $\ell_{\mathcal{G}'}(h,(x,y)) \leq \ell_{\tilde{\mathcal{G}}}(h,(x,y)) \leq \ell_{\mathcal{G}}(h,(x,y))$ and (2) $\ell_{\tilde{\mathcal{G}}}(h,(x,y))$ enjoys the uniform convergence property with sample complexity $O\left(\frac{\text{VC}(\mathcal{H})\log(\mathcal{N}_r(\mathcal{G}'_{2r},d))\ln(\frac{1}{\epsilon})+\ln(\frac{1}{\delta})}{\epsilon^2}\right)$. Let $\tilde{\mathcal{G}} \subset \mathcal{G}$ be the minimal $r$-cover of $\mathcal{G}'_{2r}$ with cardinality $\mathcal{N}_r(\mathcal{G}'_{2r},d)$. By Lemma 1.1 of Attias et al. [2021], the loss class $\mathcal{L}_{\mathcal{H}}^{\tilde{\mathcal{G}}}$ has VC dimension at most $O(\text{VC}(\mathcal{H})\log(|\tilde{\mathcal{G}}|)) = O(\text{VC}(\mathcal{H})\log(\mathcal{N}_r(\mathcal{G}'_{2r})))$, implying that $\ell_{\tilde{\mathcal{G}}}(h,(x,y))$ enjoys the uniform convergence property with the previously stated sample complexity $O\left(\frac{\text{VC}(\mathcal{H})\log(\mathcal{N}_r(\mathcal{G}'_{2r},d))\ln(\frac{1}{\epsilon})+\ln(\frac{1}{\delta})}{\epsilon^2}\right)$. Now, it remains to show that for our choice of $\tilde{\mathcal{G}}$, we have $\ell_{\mathcal{G}'}(h,(x,y)) \leq \ell_{\tilde{\mathcal{G}}}(h,(x,y)) \leq \ell_{\mathcal{G}}(h,(x,y))$. Since, $\tilde{\mathcal{G}} \subset \mathcal{G}$ ,the upperbound is trivial. Thus, we only focus on proving the lowerbound, $\ell_{\mathcal{G}'}(h,(x,y)) \leq \ell_{\tilde{\mathcal{G}}}(h,(x,y))$ for all $h \in \mathcal{H}$ and $(x,y) \in \mathcal{X} \times \mathcal{Y}$. Fix $h \in \mathcal{H}$ and $(x,y) \in \mathcal{X} \times \mathcal{Y}$. If $\ell_{\mathcal{G}'}(h,(x,y)) = 1$, then there exists a $g \in \mathcal{G}'$ such that $h(g(x)) \neq y$. Let $g$ denote one such perturbation function. By the $r$-Niceness property of $\mathcal{G}'$ with respect to $\mathcal{H}$, there must exist $B_r(g^*)$ centered at some $g^* \in \mathcal{G}$ such that $g \in B_r(g^*)$ and $h(g(x)) = h(g'(x))$ for all $g' \in B_r(g^*)$. This implies that $h(g'(x)) \neq y$ for all $g' \in B_r(g^*)$. Furthermore, since $B_{2r}(g)$ is the union of all balls of radius $r$ that contain $g$, we have that $B_r(g^*) \subset B_{2r}(g)$. From here, its not too hard to see that $B_r(g^*) \subset \mathcal{G}'_{2r}$ by definition. Finally, since $\tilde{\mathcal{G}}$ is an $r$-cover of $\mathcal{G}'_{2r}$, it must contain at least one function from $B_r(g^*)$. This completes the proof as we have shown that there exists a perturbation function $\hat{g} \in \tilde{\mathcal{G}}$ such that $h(\hat{g}(x)) \neq y$. $\qquad\square$

### D.4 $\ell_p$ balls are $r$-Nice perturbation sets for linear classifiers

In this section, we give a concrete example of a hypothesis class $\mathcal{H}$ and metric space of perturbation functions $(\mathcal{G},d)$ for which there exists an $r$-nice perturbation subset $\mathcal{G}' \subset \mathcal{G}$. Let $\mathcal{X} = \mathbb{R}^q$ and fix $r \in \mathbb{R}_{\geq 0}$. For the hypothesis class, consider the set of homogeneous halfspaces, $\mathcal{H} = \{h_w|w \in \mathbb{R}^q\}$, where $h_w(x) = w^T x$. Let $\hat{\mathcal{G}} = \{g_\delta : \delta \in \mathbb{R}^q, ||\delta||_p \leq 3r\}$ where $g_\delta(x) = x + \delta$ for all $x \in \mathcal{X}$ and consider *any* perturbation set $\mathcal{G}$ such that $\mathcal{G} \supset \hat{\mathcal{G}}$. That is, $\hat{\mathcal{G}}(x) = \{g(x) : g \in \hat{\mathcal{G}}\}$ induces a $\ell_p$ ball of radius $3r$ around $x$. We will accordingly consider the distance metric $d(g_{\delta_1}, g_{\delta_2}) = \sup_{x\in\mathcal{X}}||g_{\delta_1}(x) - g_{\delta_2}(x)||_p$. Restricted to the set $\hat{\mathcal{G}}$, this distance metric reduces to $d(g_{\delta_1}, g_{\delta_2}) = ||\delta_1-\delta_2||_p = \ell_p(\delta_1,\delta_2)$ for $g_{\delta_1}, g_{\delta_2} \in \hat{\mathcal{G}}$. Finally, consider $\mathcal{G}' = \{g_\tau : \tau \in \mathbb{R}^q, ||\tau||_p \leq r\} \subset \hat{\mathcal{G}} \subset \mathcal{G}$ which induces an $\ell_p$ ball of radius $r$ around $x$.

We will now show that $\mathcal{G}'$ is $r$-nice perturbation set with respect to $\mathcal{H}$. Let $x \in \mathcal{X}$, $h_w \in \mathcal{H}$, and $g_\tau \in \mathcal{G}'$. Let $c = h(g_\tau(x)) \in \{\pm1\}$. Consider the function $g_{\tau+\frac{crw}{||w||_p}}$. By definition, we have that $g_\tau \in B_r(g_{\tau+\frac{crw}{||w||_p}}) \subset \hat{\mathcal{G}} \subset \mathcal{G}$. To see this, observe that $||\tau + \frac{crw}{||w||_p}||_p \leq 2r$ by the triangle inequality. Finally, it remains to show that for every $g' \in B_r(g_{\tau+\frac{crw}{||w||_p}}) = \{g_{\tau+\frac{crw}{||w||_p}+\kappa}|\kappa \in \mathbb{R}^d, ||\kappa||_p \leq r\}$, $h_w(g'(x)) = h_w(g_\tau(x)) = c$. Let $c = +1$ and consider the function $g'_{\tau+\frac{rw}{||w||_p}+\kappa} \in B_r(g_{\tau+\frac{rw}{||w||_p}})$.

Note that $w^T(x + \tau + \frac{rw}{||w||_p} + \kappa) = w^T(x+\tau) + r||w||_p + w^T\kappa$. By Cauchy-Schwartz, we can lower bound $w^T\kappa \geq -||w||_p||\kappa||_p \geq -r||w||_p$. Therefore, we have that $w^T(x + \tau + \frac{rw}{||w||_p} + \kappa) \geq w^T(x + \tau) > 0$, where the last inequality comes from the fact that $+1 = c = h_w(g_\tau) = \text{sign}(w^T(x+\tau))$. Therefore, $h(g'_{\tau + \frac{rw}{||w||_p} + \kappa}(x)) = \text{sign}(w^T(x + \tau + \frac{rw}{||w||_p} + \kappa)) = \text{sign}(w^T(x+\tau)) = h(g_\tau(x))$ as desired. A similar proof holds when $c = -1$. Therefore, we have shown that $\mathcal{G}'$ is a $r$-nice perturbation set with respect to $\mathcal{H}$.

We now can use Theorem 5.4 to provide sample complexity guarantees on Tolerantly Robust PAC Learning with $\mathcal{G}'$ and $\mathcal{G}$. The main quantity of interest is $\log(\mathcal{N}_r(\mathcal{G}'_{2r}, d))$. However, note that $\mathcal{G}'_{2r} = \hat{\mathcal{G}}$. Therefore, we just need to compute $\log(\mathcal{N}_r(\hat{\mathcal{G}}, d)) = \log(\mathcal{N}_r(\{g_\delta : \delta \in \mathbb{R}^q, ||\delta||_p \leq 3r\}, d))$. However, this is equal to $\log(\mathcal{N}_r(\{\delta \in \mathbb{R}^q : ||\delta||_p \leq 3r\}, \ell_p))$ using the $\ell_p$ distance metric since $g_\delta$ maps one-to-one to $\delta$. Using standard arguments, $\log(\mathcal{N}_r(\{\delta \in \mathbb{R}^q : ||\delta||_p \leq 3r\}, \ell_p)) = \log(\mathcal{N}_{\frac{1}{3}}(\{\delta \in \mathbb{R}^q : ||\delta||_p \leq 1\}, \ell_p)) = O(q)$ (Bartlett [2013]). Thus, overall, $\mathcal{H}$ is tolerantly PAC learnable with respect to $(\mathcal{G}, \mathcal{G}')$ with sample complexity close to what one would require in the standard PAC setting.