# OpenReview forum: "On Proper Learnability between Average- and Worst-case Robustness"
_NeurIPS.cc/2023/Conference — NeurIPS 2023 poster_

### Official Review · Reviewer_dXD9 · 2023-06-13

**Soundness:** 3 good
**Presentation:** 3 good
**Contribution:** 3 good
**Rating:** 7
**Confidence:** 4

**Summary:**

This paper initiates the study of a new kind of PAC learning: probabilistically robust PAC learning.  The authors show that the finiteness of the VC dimension of the function class is not sufficient to obtain a proper learning rule in this new PAC learning setup.  However, they show that for Lipschitz losses that interpolate between the average and worst case, proper learning is possible given finite VC dimension.  They also consider the settings of adversarially robust PAC learning and tolerant PAC learning; ultimately, this results in several extensions to previous works on these topics.

**Strengths:**

**New problem setting.**  This is a new problem.  The authors consider several recent results in the robustness literature concerning robustness between the average and worst case, and they form a new definition of probabilistically robust PAC learning.  I imagine that this is a direction that other may be interested in, and therefore this constitutes an interesting contribution.

**Non-proper learnability.**  Perhaps the most interesting/surprising result here is that the finiteness of the VC dimension of the function class is not sufficient for proper $\rho$-probabilistic robust PAC learning.  This implies that both probabilistic and adversarial robustness do not easily admit proper learning rules.  This is somewhat surprising, given previous empirical results that find that probabilistic robustness does not come at the cost of degraded nominal performance with respect to an empirical risk minimizer.  Building on this, it is also interesting that there do exist interpolation schemes that do admit proper learning rules given finite VC dimension.  In this way, this paper is a first step toward characterizing a hierarchy of interpolating losses based on which ones admit proper learning rules.  It would be interesting to know whether other interpolation methods, e.g., those in [Rice et al., 2021] and [Li et al., 2020] also engender proper learning rules.

**Technical rigor.**  The paper seems technically sound to me.  The proofs in the appendix are well-structured, and the array of tools used in the appendix may be of broader interest to the community.

**Weaknesses:**

**Unevenness of the presentation.**  There's a certain unevenness about the presentation of the main results.  In general, the trend is that as one gets further into the paper, the results get harder to parse.  This doesn't seem to be a function of the complexity of the results; rather, it seems as if less space was dedicated to fully explaining the results that appear later in the paper.

For instance, the results in Section 3 are outlined in detail.  Theorem 3 tells us that there exists hypothesis classes for which $\rho$-probabilistically robust PAC learning is not possible, and the proof is sketched almost in full.  However, by the time we reach Section 5, the results are stated with less explanation.  The text becomes quite difficult to read because nearly every equation on pages 8-9 is inline.  Definitions and theorems seem to be crammed in, and the paper ends without discussing the implications of the final theorem.  I think that readibility would be greatly improved if the authors moved the proofs from Section 3 to the appendix, and (i) expanded more on the results later in the paper, (ii) broke up the texts by putting long equations on their own lines (i.e., not in-line), and (iii) adding a discussion section where the implications of the results are discussed.  In its current form, the paper ends rather abruptly, and I think point (iii) would help to ameliorate this.

**Writing and notation.**  I think that the presentation could be improved in several ways.  Here are some points that occurred to me while reading the paper:

* It would also be helpful if the authors could use equation numbers.
* It's confusing that the authors introduce VC dimension and Rademacher complexity, but a definition arguably more central to this paper -- the definition of a *proper* learning rule -- was omitted.
* ERM seems to be used to denote "empirical risk minimization" (throughout) and "empirical risk minimizer" (line 95).  I think it'd be worth picking one.
* What is $(\mathcal{X}\times\mathcal{Y})^\star$, i.e., what does the *\star* denote?
* Why is there a change in notation from $\mathcal{L}^{\mathcal{H}}$ to to $\mathcal{F}$ in Definition 2?

**Section 5.3.**  When reading, I wasn't sure how Section 5.3 fit in with the rest of the paper.  Whereas the other sections of the paper focus on proper learnability for probabilistic robustness and generalizations which interpolate between average and worst-case robustness, Section 5.3 seems to address questions that are somewhat different in spirit.  I suppose one could argue that in the tolerance setting, reducing $\mathcal{G}$ to a singleton set containing the identify function would show that this paradigm can interpolate between the standard PAC model and robust variants, but this connection feels tenuous.  Perhaps the authors could elaborate more on why this section fits in with this paper.

**Questions:**

Is it fair to say that the fact that the probabilistic robustness loss function is non-Lipschitz is the reason behind Theorem 3.1.  In other words, is Lipschitzness a necessary condition?  My understand from Section 4 is that it is sufficient, but it may not be necessary.

**Overall assessment.**  Overall, I do not see a strong reason for this paper not to be accepted.  It studies a new problem and the insights are novel and interesting.  There are some drawbacks regarding the presentation, but one imagines that these can be easily ironed out.

---

> ### Author Rebuttal · Authors · 2023-08-07
>
> We thank the reviewer for finding the results to be a surprising/interesting contribution, this setting to be interesting to others, and the array of tools used to be of broader interest to the community.
>
> **Q1**: *"Unevenness of the presentation."*
>
> A1: We agree with the reviewer and will make sure to incorporate all three recommended changes:  (i) expand more on the results later in the paper, (ii) break up the texts by putting long equations on their own lines (i.e., not in-line), and (iii) add a discussion section where the implications of the results are discussed.
>
> **Q2**: *"Writing and notation."*
>
> A2: We agree with all of the reviewer's comments and will make the recommended changes in the camera-ready version. The star in (X \times Y)^* is used to denote the space of arbitrarily long, but finite, sequences of labeled instances. The change in notation from L^H to F s a typo and will be fixed in the camera-ready version.
>
> **Q3**: *"How does Section 5.3 fit in the paper?"*
>
> A3: One central theme of the paper is identifying settings where proper learners, and more specifically,  ERM works. In Section 4, we showed that ERM works when \ell is a Lipschitz loss function of the probabilistic margin. In Section 5.1, we showed that PRERM works if you allow the learner to compete against a slightly stronger notion of probabilistic robustness. In Section 5.2, we showed that RERM works if you compare the learner's probabilistic robust risk to the best adversarially robust risk over hypothesis in H. Likewise, in Section 5.3, we identify another setting where the RERM works. Sections 5.1, 5.2, and 5.3 are further unified in the sense that they all consider a learning setting where the learner competes against a slightly stronger notion of robustness. Finally, another unifying theme throughout Section 5 is the use of Lemma 5.1, named Sandwich Uniform Convergence. Indeed, Lemma 5.1 is used to prove all results in Sections 5.1, 5.2, and 5.3.
>
> **Q4**: *"Is it fair to say that the fact that the probabilistic robustness loss function is non-Lipschitz is the reason behind Theorem 3.1. In other words, is Lipschitzness a necessary condition? My understand from Section 4 is that it is sufficient, but it may not be necessary."*
>
> A4: Lipschitzness is sufficient but, in full generality, not necessary for proper learnability.  For example, the loss function that completely ignores G and \mu and just computes the 0-1 loss is not Lipschitz, however, it is learnable via ERM. That said, among those losses that are a function of the probabilistic robust margin, it is an interesting open question to understand whether Lipschitzness is necessary for proper probabilistic robust learnability. if the loss function \ell is not a Lipschitzness function of the probabilistic robust margin, we may be able to construct a counterexample similar to the one in Lemma 3.2 by having the probabilistically robust loss class "spike" in all possible combinations across the sample while maintaining low overall complexity of the original hypothesis class. We will include a discussion of this in the camera-ready version.

---

> > ### Comment · Reviewer_dXD9 · 2023-08-14
> > **Rebuttal response**
> >
> > **Q1 and Q2:**  Great, I think that this will improve the paper.
> >
> > **Q3:**  I'm still a bit confused about this.  The title of the paper is "On Proper Learnability between Average- and Worst-case Robustness," and the argument made in the rebuttal doesn't do a lot to convince the reader that tolerant PAC learning fits within the bounds of the average-to-worst-case paradigm.  Generally, I don't quite see the connection between competing with a stronger notion of robustness and probabilistic notions of robustness.
> >
> > **Q4:** I think adding this discussion will sure up this part of the paper.
> >
> > Other than that, I don't have much to say.  I think that this paper should be accepted.

---

> > > ### Author Response · Authors · 2023-08-14
> > > **Response to Reviewer dxD9**
> > >
> > > We proved a technical lemma, termed Sandwich Uniform Convergence (SUC), to derive results on proper probabilistic robust learnability. However, we show that SUC is a general technical tool that has wider applicability by including the section on Tolerant Robust Learning. We agree that Section 5.3 is slightly tangential from the main story. We will move this Section to the Appendix and use the additional space to address reviewer feedback.

---

### Official Review · Reviewer_rPBv · 2023-06-22

**Soundness:** 3 good
**Presentation:** 3 good
**Contribution:** 3 good
**Rating:** 7
**Confidence:** 4

**Summary:**

This paper investigates the relaxations of the worst-case robust loss to make VC classes properly PAC learnable. Firstly, this paper shows that an exsiting and natural relaxation does not work. Then, the paper gives a family of robust loss relaxations that interpolate between average- and worst-case robustness. Finally, the paper studies the generalization guarantees for the adversarially robust empirical risk minimizer.

**Strengths:**

- This paper shows that an exsiting and natural relaxation does not work.
- This paper gives a family of robust loss relaxations that interpolate between average- and worst-case robustness and shows that they make the VC classes properly learnable. The results are interesting.


**Weaknesses:**

- Lack of descriptions about the high-level intuitions (please refer to the questions part).
- Some minor issues. The label space is defined as $\mathcal{Y} = \\{ -1, 1 \\}$, however, in the proof of Lemma 3.2, the paper uses $\\{ 0,1 \\}$. In Lemma 4.1, it seems that $\ell$ needs to be bounded but the paper ignores it.

**Questions:**

- Would you please show some ideas about the proof of Lemma 3.2? When considering this problem, is it the case that you first consider giving an upper bound of the VC dimension of $\mathcal{L}$ in terms of the VC dimension of $\mathcal{H}$ or the case that you directly try to construct the counterexample? Would you please show some high-level thoughts about adapting the proof of Omar to the case in this paper?
- Would you please provide some high-level insights about the construction of the counterexample in Theorem 4.3?

**Limitations:**

No.

---

> ### Author Rebuttal · Authors · 2023-08-07
>
> We thank the reviewer for finding the results in this work to be interesting.
>
> **Q1**: *"Minor issues."*
>
> A1: We thank the reviewer for pointing out these issues. We will fix them in the camera-ready version.
>
> **Q2**: *"Would you please show some ideas about the proof of Lemma 3.2? When considering this problem, is it the case that you first consider giving an upper bound of the VC dimension of L in terms of the VC dimension of H, or the case that you directly try to construct the counterexample? Would you please show some high-level thoughts about adapting the proof of Omar to the case in this paper?"*
>
> A2: We will make sure to include a proof sketch of Lemma 3.2 in the camera-ready version of the paper. In order to prove Lemma 3.2, we directly construct a counter-example where VC(H) <=1 but VC(L) >= m. We do include some high-level thoughts about the differences between our construction and Montasser et. al's construction in Lines 162-173. However, we will expand more on this in the camera-ready version. In Montasser et. al's proof, in order for a hypothesis to have an adversarial robust loss of 1 on an instance x, it was sufficient to have just one perturbation in the ball on which the hypothesis was non-robust. However, in our case, in order for a hypothesis to have a probabilistic robust loss of 1 on an instance x, we need the hypothesis to be non-robust on a sufficiently large number of perturbations in the ball. Accordingly, the main challenge/contribution in our paper is how to construct these regions of non-robustness such that the VC dimension of the probabilistic robust loss class can be made arbitrarily large, yet the VC dimension of H remains small.
>
>
> **Q3**: *"Would you please provide some high-level insights about the construction of the counterexample in Theorem 4.3?"*
>
> A3: We thank the reviewer for pointing this out. For the camera-ready version, we will include a proof sketch and some high-level insights about the construction of the example relevant to Theorem 4.3. The idea is to consider the well-known infinite VC class H = {sign(sin(wx)) : w in R }, but to pick a G and \mu, such that the expectation E[h(g(x))] is essentially constant in x for all hypothesis h in H. We provide the exact example in Appendix B.2.

---

> > ### Comment · Reviewer_rPBv · 2023-08-10
> >
> > Thank you for your response, I am looking forward to reading your revision.

---

### Official Review · Reviewer_8idw · 2023-07-06

**Soundness:** 3 good
**Presentation:** 4 excellent
**Contribution:** 3 good
**Rating:** 6
**Confidence:** 4

**Summary:**

This paper studies the proper robust learnability under relaxation of the (usual) worst-case/all powerful adversary assumption.

- The authors first show that finite VC dimension is not sufficient to enable proper learnability under the relaxation proposed by Robey et al. (2022).
- For another generalization of worst-case relaxations, the authors show that finite VC dimension enables proper robust learnability
- The authors study the "relaxed competition" setting where the hypothesis is compared to the optimal hypothesis under a slightly stronger notion of robustness, proper learning guarantees are possible


**Strengths:**

- The paper is well-written, clear and easy to follow
- I believe the topic and results are of interest to the learning theory community,
- Relaxing the worst-case analysis is well-motivated

**Weaknesses:**

1. It seems perhaps a considerable number of proofs rely on standard techniques
2. Is it a limitation / too big of a relaxation to have the adversary pick a perturbation independently of the unperturbed point $x$? (l.70-71) Unless $\mu$ can be defined for each $x$. Either way, it would be worth discussing and clarifying this point in the main body (unless I have missed this somewhere).

Overall I think even if the potential limitations pointed out above are right, the paper still offers a good contribution.

**Questions:**

Could you address point 2 above?

**Limitations:**

If point 2 in "Weaknesses" is correct, it would be worth including as a limitation of the work.

---

> ### Author Rebuttal · Authors · 2023-08-07
>
> We thank the reviewer for noting that the results in this work are of interest to the learning theory community and that relaxing the worst-case analysis is well-motivated.
>
> **Q**: *"Is it a limitation / too big of a relaxation to have the adversary pick a perturbation independently of the unperturbed point x? (l.70-71) Unless μ can be defined for each x. Either way, it would be worth discussing and clarifying this point in the main body (unless I have missed this somewhere)."*
>
> A: In our model, the measure \mu is fixed beforehand and does not depend on the unperturbed point x. We will make sure to clarify this point in the main text of the camera-ready version. Allowing the measure to depend on the unperturbed point x is an interesting future direction that lies between our model and adversarial robustness. That said, we believe that having one fixed measure is natural from a practical perspective.  In practice, both G and \mu will be picked during training time (for example Robey et al. pick \ell_infty balls and the uniform measure), and it is unclear why one would want to weight different perturbations differently for different instances. One would also need to then define a measure for each instance, which might not be computationally feasible. In addition, even when the same fixed measure \mu is used for all unperturbed points x, we show that there are natural losses where proper learning is not always possible. Lastly, we note that our model of having the measure \mu fixed beforehand can be motivated by considering a computationally lazy/bounded adversary who may not be able to define and sample from different measures. We provide this motivation in lines 68-74 of the main text. We will further discuss and clarify these points in the main text of the camera-ready version.

---

> > ### Comment · Reviewer_8idw · 2023-08-11
> > **Response**
> >
> > Thanks for the response! I am looking forward to reading the discussion and clarification of $\mu$ in the final version.

---

### Official Review · Reviewer_yAnv · 2023-07-06

**Soundness:** 4 excellent
**Presentation:** 3 good
**Contribution:** 3 good
**Rating:** 7
**Confidence:** 5

**Summary:**

This paper studies the setting of robust PAC learning to test time attacks, using a relaxed notion of robustness on average instead of robustness to the worst-case attack.

The contributions are as follows.

-Negative result: even when using the relaxed notion of robustness, improper learning is impossible. This is a stronger result from the example in the worst-case setting [Montasser et al. 2019]. Moreover, this is achieved by a natural example of $\ell_p$ balls and the uniform measure.
The intuition is that the non-Lipschitzness of the 0-1 loss enforces to use improper learning.

-Positive results:
1. When considering Lipschitz loss functions, uniform convergence hold, and as a result, ERM is sufficient for learnability.
2. Instead of relaxing the robust loss, it is possible to relax the benchmark we compare to, i.e. the best function in the class but with a smaller parameter in the probabilistic loss. This is similar to the setup of Tolerant Robust PAC Learning.

**Strengths:**

This paper provides nice contributions to the literature on robust learning, by finding natural relaxations on the robust model that allows learning similar to non-robust learning.

**Weaknesses:**

See Questions.
The writing can be improved. This paper has many good ideas, but sometimes it is hard to follow them.
Also, many relevant references from theory on robust learning are missing that should be included as related work.



**Questions:**

The description of the model might be confusing. I will explain my point of view.
In the standard setting, the set of possible perturbations is fixed and known to the learner. It's not chosen by an adversary, it's just the possible attacks the learner is aiming to protect from at test time. An adversary would just pick the set of all possible perturbations.

In this model, is the set G and measure $\mu$ being chosen at training time and known to the learner? If so, that makes sense to me and should be clear in the model.
What's a reasonable choice measure? is it representing the importance of each $g$? I think that some motivation is missing.

The connection between the average case and worst case model is through using the  $\rho$-probabilistically robust loss, maybe it should be mentioned before section 3. This is a very important explanation for defining the model of robustness on average!

Some definitions are used throughout the paper. It could improve the readability if those be numbered and referred to when used.
For example, the risk under the probabilistic robust loss (line 116) is used in section 5. It takes some time to find the definition.

Many references are missing. For example,
H-consistency bounds for surrogate loss minimizers (ICML 2022),
Multi-class H-consistency bounds (NeurIPS 2022),
Theoretically grounded loss functions and algorithms for adversarial robustness (AISTATS 2023),
Cross-Entropy Loss Functions: Theoretical Analysis and Applications (ICML 2023),
A Characterization of Semi-Supervised Adversarially Robust PAC Learnability (NeurIPS 2022),
Adversarially Robust PAC Learnability of Real-Valued Functions (ICML 2023),
On the hardness of robust classification (JMLR)
...and many more!

**Limitations:**

There are no limitations.

---

> ### Author Rebuttal · Authors · 2023-08-07
>
> We thank the reviewer for finding that this paper provides a nice contribution to the literature on robust learning.
>
> **Q1**:  *"In this model, is the set G and measure μ being chosen at training time and known to the learner?"*
>
> A1: Yes, the set G and the measure \mu are chosen at training time and known to the learner. Moreover, the same G and measure \mu are used to evaluate the model at test time. We will make this more clear in the main text of the camera-ready version.
>
> **Q2**: *"What's a reasonable choice measure? is it representing the importance of each g?"*
>
> A2: If G encodes a \ell_p ball, then a reasonable choice of measure \mu over G could be the uniform measure. In this case, this measure would encode the idea that every perturbation is equally important. For example, Robey et al. use \ell_infty balls with the uniform measure in their experiments training probabilistically robust neural networks. Another reasonable choice of measure \mu could be one whose mass concentrates at the center of the ball but decays radially as you move out towards the edge of the ball. This measure would encode the idea that perturbations near the center of the ball are more important than those further out. We will make sure to include this motivation/intuition in the main text of the camera-ready version.
>
> **Q3**: *"The connection between the average case and worst case model is through using the  ρ-probabilistically robust loss, maybe it should be mentioned before section 3."*
>
> A3: We thank the reviewer for this comment and will make sure to include a discussion of this connection before Section 3 in the camera-ready version.
>
> **Q4**: *"Some definitions are used throughout the paper. It could improve the readability if those be numbered and referred to when used."*
>
> A4: We agree with the reviewer and will make the recommended changes in the camera-ready version.
>
> **Q5**: *"Many references are missing"*
>
> A5: We thank the reviewer for pointing out these missing references. We will make sure to discuss these works and reference them in the camera-ready version.

---

### Decision · Program_Chairs · 2023-09-21

**Decision:**

Accept (poster)

**Comment:**

The paper makes (theoretical) contributions to the problem of proper learning under test-time adversarial perturbations. They show that a (previously proposed) natural relaxation of the problem is not enough for proper learning, but another relaxation can be sufficient.

The reviewers are positive about this paper and its contributions. Some minor shortcomings in terms of the presentation and missing references are expected to be addressed in the camera ready.